# A fast and scalable framework for large-scale and ultrahigh-dimensional sparse regression with application to the UK Biobank

**Junyang Qian**[1], **Yosuke Tanigawa**[2], **Wenfei Du**[1], **Matthew Aguirre**[2], **Chris Chang**[3], **Robert Tibshirani**[1,2], **Manuel A. Rivas**[2], **Trevor Hastie**[1,2]*

**1** Department of Statistics, Stanford University, Stanford, CA, United States of America, **2** Department of Biomedical Data Science, Stanford University, Stanford, CA, United States of America, **3** Grail, Inc., Menlo Park, CA, United States of America

* hastie@stanford.edu

**Data Availability Statement:** The analyses presented in this study were based on data accessed through the UK Biobank: http://www.ukbiobank.ac.uk. Experiment scripts are available

## Abstract

The UK Biobank is a very large, prospective population-based cohort study across the United Kingdom. It provides unprecedented opportunities for researchers to investigate the relationship between genotypic information and phenotypes of interest. Multiple regression methods, compared with genome-wide association studies (GWAS), have already been showed to greatly improve the prediction performance for a variety of phenotypes. In the high-dimensional settings, the lasso, since its first proposal in statistics, has been proved to be an effective method for simultaneous variable selection and estimation. However, the large-scale and ultrahigh dimension seen in the UK Biobank pose new challenges for applying the lasso method, as many existing algorithms and their implementations are not scalable to large applications. In this paper, we propose a computational framework called batch screening iterative lasso (BASIL) that can take advantage of any existing lasso solver and easily build a scalable solution for very large data, including those that are larger than the memory size. We introduce **snpnet**, an R package that implements the proposed algorithm on top of **glmnet** and optimizes for single nucleotide polymorphism (SNP) datasets. It currently supports $\ell_1$-penalized linear model, logistic regression, Cox model, and also extends to the elastic net with $\ell_1/\ell_2$ penalty. We demonstrate results on the UK Biobank dataset, where we achieve competitive predictive performance for all four phenotypes considered (height, body mass index, asthma, high cholesterol) using only a small fraction of the variants compared with other established polygenic risk score methods.

## Author summary

With the advent and evolution of large-scale and comprehensive biobanks, there come up unprecedented opportunities for researchers to further uncover the complex landscape of human genetics. One major direction that attracts long-standing interest is the investigation of the relationships between genotypes and phenotypes. This includes but doesn't limit to the identification of genotypes that are significantly associated with the

on GitHub (https://github.com/junyangq/scripts_snpnet_paper).

**Funding:** This research has been conducted using the UK Biobank Resource under application number 24983, "Generating effective therapeutic hypotheses from genomic and hospital linkage data" (http://www.ukbiobank.ac.uk/wp-content/uploads/2017/06/24983-Dr-Manuel-Rivas.pdf). This work was supported by National Human Genome Research Institute (NHGRI) of the National Institutes of Health (NIH) under awards R01HG010140. This work was funded in part by the Two Sigma Graduate Fellowship (to J.Q.), Funai Overseas Scholarship from Funai Foundation for Information Technology and the Stanford University School of Medicine (to Y.T.). NIH grants 5U01HG009080 (to M.A.R.), 5R01EB001988-16 (to R.T.), 5R01EB 001988-21 (to T.H.). NSF grants 19DMS1208164 (to R.T.) and DMS-1407548 (to T. H.). The content is solely the responsibility of the authors and does not necessarily represent the official views of the funders; funders had no role in study design, data collection and analysis, decision to publish, or preparation of the manuscript.

**Competing interests:** I have read the journal's policy and the authors of this manuscript have the following competing interests: Chris Chang is an employee of Grail, Inc.

phenotypes, and the prediction of phenotypic values based on the genotypic information. Genome-wide association studies (GWAS) is a very powerful and widely used framework for the former task, having produced a number of very impactful discoveries. However, when it comes to the latter, its performance is fairly limited by the univariate nature. To address this, multiple regression methods have been suggested to fill in the gap. That said, challenges emerge as the dimension and the size of datasets both become large nowadays. In this paper, we present a novel computational framework that enables us to solve efficiently the entire lasso or elastic-net solution path on large-scale and ultrahigh-dimensional data, and therefore make simultaneous variable selection and prediction. Our approach can build on any existing lasso solver for small or moderate-sized problems, scale it up to a big-data solution, and incorporate other extensions easily. We provide a package **snpnet** that extends the **glmnet** package in R and optimizes for large phenotype-genotype data. On the UK Biobank, we observe competitive prediction performance of the lasso and the elastic-net for all four phenotypes considered from the UK Biobank. That said, the scope of our approach goes beyond genetic studies. It can be applied to general sparse regression problems and build scalable solution for a variety of distribution families based on existing solvers.

## Introduction

The past two decades have witnessed rapid growth in the amount of data available to us. Many areas such as genomics, neuroscience, economics and Internet services are producing big datasets that have high dimension, large sample size, or both. A variety of statistical methods and computing tools have been developed to accommodate this change. See, for example, [1–5] and the references therein for more details.

In high-dimensional regression problems, we have a large number of predictors, and it is likely that only a subset of them have a relationship with the response and will be useful for prediction. Identifying such a subset is desirable for both scientific interests and the ability to predict outcomes in the future. The lasso [6] is a widely used and effective method for simultaneous estimation and variable selection. Given a continuous response $y \in \mathbb{R}^n$ and a model matrix $X \in \mathbb{R}^{n \times p}$, it solves the following regularized regression problem.

$$\hat{\beta}(\lambda) = \operatorname*{argmin}_{\beta \in \mathbb{R}^p} \frac{1}{2n} \|y - X\beta\|_2^2 + \lambda\|\beta\|_1, \tag{1}$$

where $\|x\|_q = \left(\sum_{i=1}^n |x_i|^q\right)^{1/q}$ is the vector $\ell_q$ norm of $x \in \mathbb{R}^n$ and $\lambda \geq 0$ is the tuning parameter. The $\ell_1$ penalty on $\beta$ allows for selection as well as estimation. Normally there is an unpenalized intercept in the model, but for ease of presentation we leave it out, or we may assume that both $X$ and $y$ have been centered with mean 0. One typically solves the entire lasso solution path over a grid of $\lambda$ values $\lambda_1 \geq \lambda_2 \cdots \geq \lambda_L$ and chooses the best $\lambda$ by cross-validation or by predictive performance on an independent validation set. In R [7], several packages, such as **glmnet** [8] and **ncvreg** [9], provide efficient procedures to obtain the solution path for the Gaussian model (1), and for other generalized linear models with the residual sum of squared replaced by the negative log-likelihood of the corresponding model. Among them, **glmnet**, equipped with highly optimized Fortran subroutines, is widely considered the fastest off-the-shelf lasso solver. It can, for example, fit a sequence of 100 logistic regression models on a sparse dataset with 54 million samples and 7 million predictors within only 2 hours [10].

However, as the data become increasingly large, many existing methods and tools may not be able to serve the need, especially if the size exceeds the memory size. Most packages, including the ones mentioned above, assume that the data or at least its sparse representation can be fully loaded in memory and that the remaining memory is sufficient to hold other intermediate results. This becomes a real bottleneck for big datasets. For example, in our motivating application, the UK Biobank genotypes and phenotypes dataset [11] contains about 500,000 individuals and more than 800,000 genotyped single nucleotide polymorphisms (SNPs) and small indel measurements per person. This provides unprecedented opportunities to explore more comprehensive genotypic relationships with phenotypes of interest. For polygenic traits such as height and body mass index (BMI), specific variants discovered by genome-wide association studies (GWAS) used to explain only a small proportion of the estimated heritability [12], an upper bound of the proportion of phenotypic variance explained by the genetic components. While GWAS with larger sample size on the UK Biobank can be used to detect more SNPs and rare variants, their prediction performance is fairly limited by univariate models. It is very interesting to see if full-scale multiple regression methods such as the lasso or elastic-net can improve the prediction performance and simultaneously select relevant variants for the phenotypes. That being said, the computational challenges are two fold. First is the memory bound. Even though each bi-allelic SNP value can be represented by only two bits and the **PLINK** software and its `bed/pgen` format [13, 14] stores such SNP datasets in a binary compressed format, statistical packages such as **glmnet** and **ncvreg** require that the data be loaded in memory in a normal double-precision format. Given its sample size and dimension, the genotype matrix itself will take up around one terabyte of space, which may well exceed the size of the memory available and is infeasible for the packages. Second is the efficiency bound. For a larger-than-RAM dataset, it has to sit on the disk and we may only read part of it into the memory. In such scenario, the overall efficiency of the algorithm is not only determined by the number of basic arithmetic operations but also the disk I/O—data transfer between the memory and the disk—an operation several magnitudes slower than in-memory operations.

In this paper, we propose an efficient and scalable meta algorithm for the lasso called Batch Screening Iterative Lasso (BASIL) that is applicable to larger-than-RAM datasets and designed to tackle the memory and efficiency bound. It computes the entire lasso path and can easily build on any existing package to make it a scalable solution. As the name suggests, it is done in an iterative fashion on an adaptively screened subset of variables. At each iteration, we exploit an efficient, parallelizable screening operation to significantly reduce the problem to one of manageable size, solve the resulting smaller lasso problem, and then reconstruct and validate a full solution through another efficient, parallelizable step. In other words, the iterations have a screen-solve-check substructure. That being said, it is the goal and also the guarantee of the BASIL algorithm that the final solution exactly solves the full lasso problem (1) rather than any approximation, even if the intermediate steps work repeatedly on subsets of variables.

The screen-solve-check substructure is inspired by [15] and especially the proposed strong rules. The strong rules state: assume $\hat{\beta}(\lambda_{k-1})$ is the lasso solution in (1) at $\lambda_{k-1}$, then the $j$th predictor is discarded at $\lambda_k$ if

$$|x_j^\top(y - X\hat{\beta}(\lambda_{k-1}))| < \lambda_k - (\lambda_{k-1} - \lambda_k). \tag{2}$$

The key idea is that the inner product above is almost "non-expansive" in $\lambda$ and that the lasso solution is characterized equivalently by the Karush-Kuhn-Tucker (KKT) condition [16]. For

the lasso, the KKT condition states that $\hat{\beta} \in \mathbb{R}^p$ is a solution to (1) if for all $1 \leq j \leq p$,

$$\frac{1}{n} \cdot x_j^\top (y - X\hat{\beta}) \begin{cases} = \lambda \cdot \text{sign}(\hat{\beta}_j), & \text{if } \hat{\beta}_j \neq 0, \\ \leq \lambda, & \text{if } \hat{\beta}_j = 0. \end{cases} \quad (3)$$

The KKT condition suggests that the variables discarded based on the strong rules would have coefficient 0 at the next $\lambda_k$. The checking step comes into play because this is not a guarantee. The strong rules can fail, though failures occur rarely when $p > n$. In any case, the KKT condition will be checked to see if the coefficients of the left-out variables are indeed 0 at $\lambda_k$. If the check fails, we add in the violated variables and repeat the process. Otherwise, we successfully reconstruct a full solution and move to the next $\lambda$. This is the iterative algorithm proposed by these authors and has been implemented efficiently into the **glmnet** package.

The BASIL algorithm proceeds in a similar way but is designed to optimize for datasets that are too big to fit into the memory. Considering the fact that screening and KKT check need to scan through the entire data and are thus costly in the disk Input/Output (I/O) operations, we attempt to do batch screening and solve *a series of* models (at different $\lambda$ values) in each iteration, where a single sweep over the full data would suffice. Followed by a checking step, we can obtain the lasso solution for multiple $\lambda$'s in one iteration. This can effectively reduce the total number of iterations needed to compute the full solution path and thus reduce the expensive disk read operations that often cause significant delay in the computation. The process is illustrated in Fig 1 and will be detailed in the next section.

## Results

### Overview of the BASIL algorithm

For convenience, we first introduce some notation. Let $\Omega = \{1, 2, \ldots, p\}$ be the universe of variable indices. For $1 \leq \ell \leq L$, let $\hat{\beta}(\lambda_\ell)$ be the lasso solution at $\lambda = \lambda_\ell$, and $\mathcal{A}(\lambda_\ell) = \{1 \leq j \leq p : \hat{\beta}_j(\lambda_\ell) \neq 0\}$ be the active set. When $X$ is a matrix, we use $X_{\mathcal{S}}$ to represent the submatrix including only columns indexed by $\mathcal{S}$. Similarly when $\beta$ is a vector, $\beta_{\mathcal{S}}$ represents the subvector including only elements indexed by $\mathcal{S}$. Given any two vectors $a, b \in \mathbb{R}^n$, the dot product or inner product can be written as $a^\top b = \langle a, b \rangle = \sum_{i=1}^n a_i b_i$. Throughout the paper, we use predictors, features, variables and variants interchangeably. We use the strong set to refer to the screened subset of variables on which the lasso fit is computed at each iteration, and the active set to refer to the subset of variables with nonzero lasso coefficients.

Remember that our goal is to compute the exact lasso solution (1) for larger-than-RAM datasets over a grid of regularization parameters $\lambda_1 > \lambda_2 > \cdots > \lambda_L \geq 0$. We describe the procedure for the Gaussian family in this section and discuss extension to general problems in the next. A common choice is $L = 100$ and $\lambda_1 = \max_{1 \leq j \leq p} |x_j^\top r^{(0)}|/n$, the largest $\lambda$ at which the estimated coefficients start to deviate from zero. Here $r^{(0)} = y$ if we do not include an intercept term and $r^{(0)} = y - \bar{y}$ if we do. In general, $r^{(0)}$ is the residual of regressing $y$ on the unpenalized variables, if any. The other $\lambda$'s can be determined, for example, by an equally spaced array on the log scale. The solution path is found iteratively with a screening-solving-checking substructure similar to the one proposed in [15]. Designed for large-scale and ultrahigh-dimensional data, the BASIL algorithm can be viewed as a batch version of the strong rules. At each iteration we attempt to find valid lasso solution for *multiple $\lambda$*

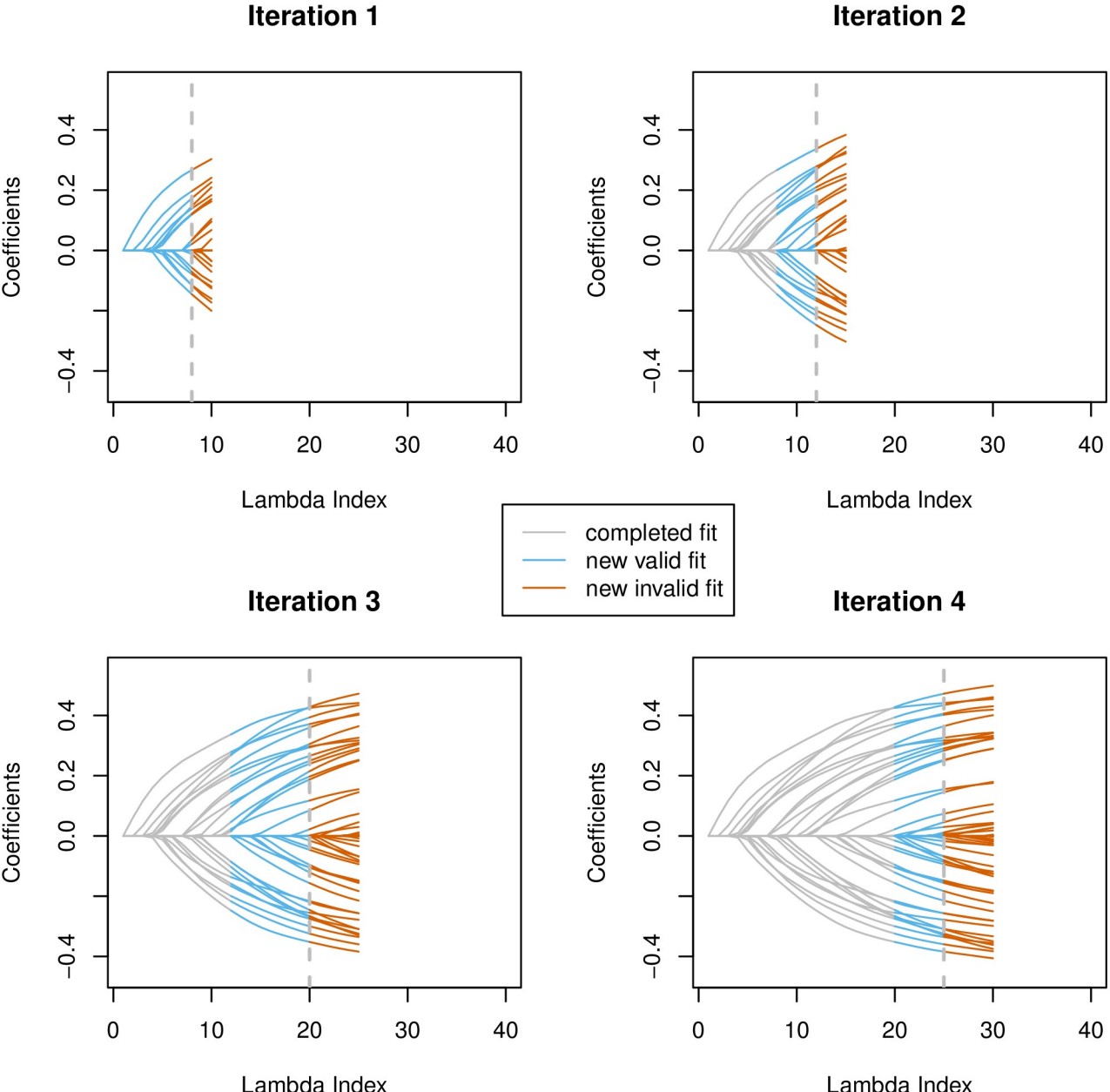

**Fig 1. The lasso coefficient profile that shows the progression of the BASIL algorithm.** The horizontal axis represents the index of lambda values, 1 $\leq \ell \leq L$, which correspond to the sequence of the regularization parameters, $\lambda_1 > \lambda_2 > \cdots > \lambda_L$. The previously finished part of the path is colored grey, the newly completed and verified is in sky blue, and the part that is newly computed but failed the verification is colored orange. The largest lambda index with the verified model is highlighted with vertical dotted gray line.

values on the path and thus reduce the burden of disk reads of the big dataset. Specifically, as summarized in Algorithm 1, we start with an empty strong set $\mathcal{S}^{(0)} = \varnothing$ and active set $\mathcal{A}^{(0)} = \varnothing$. Each of the following iterations consists of three steps: screening, fitting and checking.

**Algorithm 1** BASIL for the Gaussian Model

1: **Initialization**: active set $\mathcal{A}^{(0)} = \varnothing$, initial residual $r^{(0)}$ (with respect to the intercept or other unpenalized variables) at $\lambda_1 = \lambda_{\max}$, a short list of initial parameters $\Lambda^{(0)} = \{\lambda_1, \ldots, \lambda_{L^{(0)}}\}$.

2: **for** $k$ = 0 to $K$ **do**

3: **Screening**: for each $1 \le j \le p$, compute inner product with current residual $c_j^{(k)} = \langle x_j, r^{(k)} \rangle$. Construct the strong set

$$\mathcal{S}^{(k)} = \mathcal{A}^{(k)} \cup \mathcal{E}_M^{(k)},$$

 where $\mathcal{E}_M^{(k)}$ is the set of $M$ variables in $\Omega \backslash \mathcal{A}^{(k)}$ with largest $|c^{(k)}|$.

4: **Fitting**: for all $\lambda \in \Lambda^{(k)}$, solve the lasso only on the strong set $\mathcal{S}^{(k)}$, and find the coefficients $\hat{\beta}^{(k)}(\lambda)$ and the residuals $r^{(k)}(\lambda)$.

5: **Checking**: search for the smallest $\lambda$ such that the KKT conditions are satisfied, i.e.,

$$\bar{\lambda}^{(k)} = \min\{\lambda \in \Lambda^{(k)} : \max_{j \in \Omega \backslash \mathcal{S}^{(k)}} (1/n)|x_j^\top r^{(k)}(\lambda)| < \lambda\}.$$

 For empty set, we define $\bar{\lambda}^{(k)}$ to be the immediate previous $\lambda$ to $\Lambda^{(k)}$ butincrement $M$ by $\Delta M$. Let the current active set $\mathcal{A}^{(k+1)}$ and residuals $r^{(k+1)}$ defined by the solution at $\bar{\lambda}^{(k)}$. Define the next parameter list $\Lambda^{(k+1)} = \{\lambda \in \Lambda^{(k)} : \lambda < \bar{\lambda}^{(k)}\}$. Extend this list if it consists of too few elements. For $\lambda \in \Lambda^{(k)} \backslash \Lambda^{(k+1)}$, we obtain exact lasso solutions for the full problem:

$$\hat{\beta}_{\mathcal{S}^{(k)}}(\lambda) = \hat{\beta}^{(k)}(\lambda), \quad \hat{\beta}_{\Omega \backslash \mathcal{S}^{(k)}}(\lambda) = 0.$$

6: **end for**

In the screening step, an updated strong set is found as the candidate for the subsequent fitting. Suppose that so far (valid) lasso solutions have been found for $\lambda_1, \ldots, \lambda_\ell$ but not for $\lambda_{\ell+1}$. The new set will be based on the lasso solution at $\lambda_\ell$. In particular, we will select the top $M$ variables with largest absolute inner products $|\langle x_j, y - X\hat{\beta}(\lambda_\ell)\rangle|$. They are the variables that are most likely to be active in the lasso model for the next $\lambda$ values. In addition, we include the ever-active variables at $\lambda_1, \ldots, \lambda_\ell$ because they have been "important" variables and might continue to be important at a later stage.

In the fitting step, the lasso is fit on the updated strong set for the next $\lambda$ values $\lambda_{\ell+1}, \ldots, \lambda_{\ell'}$. Here $\ell'$ is often smaller than $L$ because we do not have to solve for all of the remaining $\lambda$ values on this strong set. The full lasso solutions at much smaller $\lambda$'s are very likely to have active variables outside of the current strong set. In other words even if we were to compute solutions for those very small $\lambda$ values on the current strong set, they would probably fail the KKT test. These $\lambda$'s are left to later iterations when the strong set is expanded.

In the checking step, we check if the newly obtained solutions on the strong set can be valid part of the full solutions by evaluating the KKT condition. Given a solution $\hat{\beta}_{\mathcal{S}} \in \mathbb{R}^{|\mathcal{S}|}$ to the sub-problem at $\lambda$, if we can verify for every left-out variable $j$ that $(1/n)|\langle x_j, y - X_{\mathcal{S}}\hat{\beta}_{\mathcal{S}}\rangle| < \lambda$, we can then safely set their coefficients to 0. The full lasso solution $\hat{\beta}(\lambda) \in \mathbb{R}^p$ is then assembled by letting $\hat{\beta}_{\mathcal{S}}(\lambda) = \hat{\beta}_{\mathcal{S}}$ and $\hat{\beta}_{\Omega \backslash \mathcal{S}}(\lambda) = 0$. We look for the $\lambda$ value prior to the one that causes the

first failure down the $\lambda$ sequence and use its residual as the basis for the next screening. Nevertheless, there is still chance that none of the solutions on the current strong set passes the KKT check for the $\lambda$ subsequence considered in this iterations. That suggests the number of previously added variables in the current iteration was not sufficient. In this case, we are unable to move forward along the $\lambda$ sequence, but will fall back to the $\lambda$ value where the strong set was last updated and include $\Delta M$ more variables based on the sorted absolute inner product.

The three steps above can be applied repeatedly to roll out the complete lasso solution path for the original problem. However, if our goal is choosing the best model along the path, we can stop fitting once an optimal model is found evidenced by the performance on a validation set. At a high level, we run the iterative procedure on the training data, monitor the error on the validation set, and stop when the model starts to overfit, or in other words, when the validation error shows a clear upward trend.

## Extension to general problems

It is straightforward to extend the algorithm from the Gaussian case to more general problems. In fact, the only changes we need to make are the screening step and the strong set update step. Wherever the strong rules can be applied, we have a corresponding version of the iterative algorithm. In [15], the general problem is

$$\hat{\beta}(\lambda) = \underset{\beta \in \mathbb{R}^p}{\mathrm{argmin}}\ f(\beta) + \lambda \sum_{j=1}^{r} c_j \|\beta_j\|_{p_j}, \tag{4}$$

where $f$ is a convex differentiable function, and for all $1 \le j \le r$, a separate penalty factor $c_j \ge 0$, $p_j \ge 1$, and $\beta_j$ can be a scalar or vector whose $\ell_{p_j}$-norm is represented by $\|\beta_j\|_{p_j}$. If all $c_j$'s are positive, we can derive that the starting value of the $\lambda$ sequence (i.e. the minimum value of $\lambda$ such that all coefficients are 0) is $\lambda_{\max} = \max_{1 \le j \le r} \|\nabla_j f(0)\|_{q_j}/c_j$. The general strong rule discards predictor $j$ if

$$\|\nabla_j f(\hat{\beta}(\lambda_{k-1}))\|_{q_j} < c_j(2\lambda_k - \lambda_{k-1}), \tag{5}$$

where $1/p_j + 1/q_j = 1$. Hence, our algorithm can adapt and screen by choosing variables with large values of $\|\nabla_j f(\hat{\beta}(\lambda_{k-1}))\|_{q_j}$ that are not in the current active set. We expand in more detail two important applications of the general rule: logistic regression and Cox's proportional hazards model in survival analysis.

## Logistic regression

In the lasso penalized logistic regression [17] where the observed outcome $y \in \{0, 1\}^n$, the convex differential function in (4) is

$$f(\beta) = -\frac{1}{n}\sum_{i=1}^{n}(y_i \log p_i + (1 - y_i) \log(1 - p_i)).$$

where $p_i = 1/(1 + \exp(-x_i^\top \beta))$ for all $1 \le i \le n$. The rule in (5) is reduced to

$$|x_j^\top(y - \hat{p}(\lambda_{k-1}))| < \lambda_k - (\lambda_{k-1} - \lambda_k),$$

where $\hat{p}(\lambda_{k-1})$ is the predicted probabilities at $\lambda = \lambda_{k-1}$. Similar to the Gaussian case, we can still fit relaxed lasso [24] and allow adjustment covariates in the model to adjust for confounding effect.

## Cox's proportional hazards model

In the usual survival analysis framework, for each sample, in addition to the predictors $x_i \in \mathbb{R}^p$ and the observed time $y_i$, there is an associated right-censoring indicator $\delta_i \in \{0, 1\}$ such that $\delta_i$ = 0 if failure and $\delta_i$ = 1 if right-censored. Let $t_1 < t_2 < \ldots < t_m$ be the increasing list of unique failure times, and $j(i)$ denote the index of the observation failing at time $t_i$. The Cox's proportional hazards model [18] assumes the hazard for the $i$th individual as $h_i(t) = h_0(t) \exp(x_i^\top \beta)$ where $h_0(t)$ is a shared baseline hazard at time $t$. We can let $f(\beta)$ be the negative log partial likelihood in (4) and screen based on its gradient at the most recent lasso solution as suggested in (5). In particular,

$$f(\beta) = -\frac{1}{m} \sum_{i=1}^{m} \left( x_{j(i)}^\top \beta - \log \left( \sum_{j \in R_i} \exp(x_j^\top \beta) \right) \right),$$

where $R_i$ is the set of indices $j$ with $y_j \geq t_i$ (those at risk at time $t_i$). We can derive the associated rule based on (5) and thus the survival BASIL algorithm. Further discussion and comprehensive experiments are included in a follow-up paper [19].

## Extension to the elastic net

Our discussion so far focuses solely on the lasso penalty, which aims to achieve a rather sparse set of linear coefficients. In spite of good performance in many high-dimensional settings, it has limitations. For example, when there is a group of highly correlated variables, the lasso will often pick out one of them and ignore the others. This poses some hardness in interpretation. Also, under high-correlation structure like that, it has been empirically observed that when the predictors are highly correlated, the ridge can often outperform the lasso [6].

The elastic net, proposed in [20], extends the lasso and tries to find a sweet spot between the lasso and the ridge penalty. It can capture the grouping effect of highly correlated variables and sometimes perform better than both methods especially when the number of variables is much larger than the number of samples. In particular, instead of imposing the $\ell_1$ penalty, the elastic net solves the following regularized regression problem.

$$\hat{\beta}(\lambda) = \underset{\beta \in \mathbb{R}^p}{\operatorname{argmin}} \; f(\beta) + \lambda(\alpha \|\beta\|_1 + (1 - \alpha)\|\beta\|_2^2/2), \tag{6}$$

where the mixing parameter $\alpha \in [0, 1]$ determines the proportion of lasso and ridge in the penalty term.

It is straightforward to adapt the BASIL procedure to the elastic net. It follows from the gradient motivation of the strong rules and KKT condition of convex optimization. We take the Gaussian family as an example. The others are similar. In the screening step, it is easy to derive that we can still rank *among the currently inactive variables* on their absolute inner product with the residual $|x_j^\top (y - X\hat{\beta}(\lambda_{k-1}))|$ to determine the next candidate set. In the checking step, to verify that all the left-out variables indeed have zero coefficients, we need to make sure that $(1/n)|x_j^\top (y - X\hat{\beta}(\lambda_{k-1}))| \leq \lambda \alpha$ holds for all such variables. It turns out that in our UK Biobank applications, the elastic-net results (after selection of $\alpha$ and $\lambda$ on the validation set) do not differ significantly from the lasso results, which will be immediately seen in the next section.

## UK Biobank analysis

We describe a real-data application on the UK Biobank that in fact motivates our development of the BASIL algorithm.

The UK Biobank [11] is a very large, prospective population-based cohort study with individuals collected from multiple sites across the United Kingdom. It contains extensive genotypic and phenotypic detail such as genomewide genotyping, questionnaires and physical measures for a wide range of health-related outcomes for over 500,000 participants, who were aged 40-69 years when recruited in 2006-2010. In this study, we are interested in the relationship between an individual's genotype and his/her phenotypic outcome. While GWAS focus on identifying SNPs that may be marginally associated with the outcome using univariate tests, we would like to find relevant SNPs in a multivariate prediction model using the lasso. A recent study [21] fits the lasso on a subset of the variables after one-shot univariate $p$-value screening and suggests improvement in explaining the variation in the phenotypes. However, the left-out variants with relatively weak marginal association may still provide additional predictive power in a multiple regression environment. The BASIL algorithm enables us to fit the lasso model at full scale and gives further improvement in the explained variance over the alternative models considered.

We focused on 337,199 White British unrelated individuals out of the full set of over 500,000 from the UK Biobank dataset [11] that satisfy the same set of population stratification criteria as in [22]. The dataset is partitioned randomly into training (60%), validation (20%) and test (20%) subsets (Methods). Each individual has up to 805,426 measured variants, and each variant is encoded by one of the four levels where 0 corresponds to homozygous major alleles, 1 to heterozygous alleles, 2 to homozygous minor alleles and NA to a missing genotype. In addition, we have available covariates such as age, sex, and fortypre-computed principal components of the SNP matrix.

To evaluate the predictive performance for quantitative response, we use a common measure R-squared ($R^2$). Given a linear estimator $\hat{\beta}$ and data $(y, X)$, it is defined as

$$R^2 = 1 - \frac{\|y - X\hat{\beta}\|_2^2}{\|y - \bar{y}\|_2^2}.$$

We evaluate this criteria for all the training, validation and test sets. For a dichotomous response, misclassification error could be used but it would depend on the calibration. Instead the receiver operating characteristic (ROC) curve provides more information and illustrates the tradeoff between true positive and false positive rates under different thresholds. The AUC computes the area under the ROC curve—a larger value indicates a generally better classifier. Therefore, we will evaluate AUCs on the training, validation and test sets for dichotomous responses.

We compare the performance of the lasso with related methods to have a sense of the contribution of different components. Starting from the baseline, we fit a linear model that includes only age and sex (Model 1 in the tables below), and then one that includes additionally the top 10 principal components (Model 2). These are the adjustment covariates used in our main lasso fitting and we use these two models to highlight the contribution of the SNP information over and above that of age, sex and the top 10 PCs. In addition, the strongest univariate model is also evaluated (Model 3). This includes the 12 adjustment covariates together with the single SNP that is most correlated with the outcome after adjustment. Toward multivariate models, we first compare with a univariate method with some multivariate flavor (Models 4 and 5). We select a subset of the $K$ most marginally significant variants (after adjusting for the covariates), construct a new variable by linearly combining these variants using their univariate coefficients, and fit an ordinary least squares (OLS) on the new variable together with the adjustment variables. It is similar to a one-step partial least squares [23] with $p$-value based truncation. We take $K = 10, 000$ and $100, 000$ in the experiments. We further compare with a hierarchical sequence of multivariate models where each is fit on a subset of the most significant SNPs. In particular, the $\ell$-th model selects $\ell \times 1000$ SNPs with the smallest

univariate $p$-values, and a multivariate linear or logistic regression is fit on those variants jointly. The sequence of models are evaluated on the validation set, and the one with the smallest validation error is chosen. We call this method Sequential LR or SeqLR (Model 6) for convenience in the rest of the paper. As a byproduct of the lasso, the relaxed lasso [24] fits a debiased model by refitting an OLS on the variables selected by the lasso. This can potentially recover some of the bias introduced by lasso shrinkage. For the elastic-net, we fit separate solution paths with varying $\lambda$'s at $\alpha = 0.1, 0.5, 0.9$, and evaluate their performance ($R^2$ or AUC) on the validation set. The best pair of hyperparameters ($\alpha, \lambda$) is selected and the corresponding test performance is reported.

There are thousands of measured phenotypes in the dataset. For demonstration purpose, we analyze four phenotypes that are known to be highly or moderately heritable and polygenic. For these complex traits, univariate studies may not find SNPs with smaller effects, but the lasso model may include them and predict the phenotype better. We look at two quantitative traits: standing height and body mass index (BMI), which are defined as a non-NA median of up to 3 measurements [25], and two qualitative traits: asthma and high cholesterol (HC) [22].

We first summarize the test performance of the methods above in Fig 2. The lasso and elastic net show significant improvement in test $R^2$ and AUC over the others. Details of the model for height are given in the next section and for the other phenotypes (BMI, asthma and high cholesterol) in Supporting Information. A comparison of the univariate $p$-values and the lasso coefficients for all these traits is shown in the form of Manhattan plots and coefficient plots in the Supporting Information.

There are many other well-established methods for constructing the polygenic risk scores from large-scale cohorts. Among them, we compare with ridge regression, pruning and thresholding (P + T), clumping, and summary statistics-based Bayesian regression methods such as PRS-CS [26] and SBayesR [27]. Ridge regression, also known as BLUP in the quantitative genetics field, fits a multiple linear regression model with $\ell_2$-penalty. It is a special case of the elastic-net with $\alpha = 0$ in (6). While it is simple and has been widely used in a variety of prediction tasks, the fact that the resulting model always includes all the variables can pose great computational challenge (for example, memory) in large-scale problems. In our experiments, the size of the data prevents us from doing exact, full-scale ridge regression. Instead, we approximate its performance by fitting the elastic-net with very small $\alpha = 10^{-3}$, which can be easily handled by our **snpnet** package. For P + T, we first identified LD independent set of variants using `--indep-pairwise 50 5.5` subcommand in PLINK2.0. We subsequently applied univariate genome-wide association analysis (`--glm firth-fallback`), focused on the LD independent variants, imposed the different p-value thresholds ($1 \times 10^{-3}$, $1 \times 10^{-4}$, and $1 \times 10^{-5}$), and extracted the univariate BETAs for the remaining variants to construct PRS [13]. For clumping, we applied `--clump` subcommand to the GWAS summary statistics with a varying ($1 \times 10^{-3}$, $1 \times 10^{-4}$, and $1 \times 10^{-5}$) p-value threshold (`--clump-p1`), and extracted the univariate BETAs for the identified lead SNPs [13]. For each of those two methods, we computed the PRS for each individual and fit an additional model consisting of covariates and the genotype PRS to report the predictive performance of the model consisting of both the genetic features and covariates. For PRS-CS, we first characterized the GWAS summary statistics using the combined set of training and validation set ($n = 269, 927$) with age, sex, and the top 10 PCs as covariates using PLINK v2.00a3LM (9 Apr 2020) [13, 28]. Using the LD reference dataset precomputed for the European Ancestry using the 1000 genome samples (https://github.com/getian107/PRScs), we applied PRS-CS with the default option. We took the posterior effect size estimates and computed the polygenic risk scores using PLINK2's `--score` subcommand [13]. For SBayesR, we computed the sparse LD matrix using the combined set of training and validation set individuals ($n = 269, 927$) using the `-- make-`

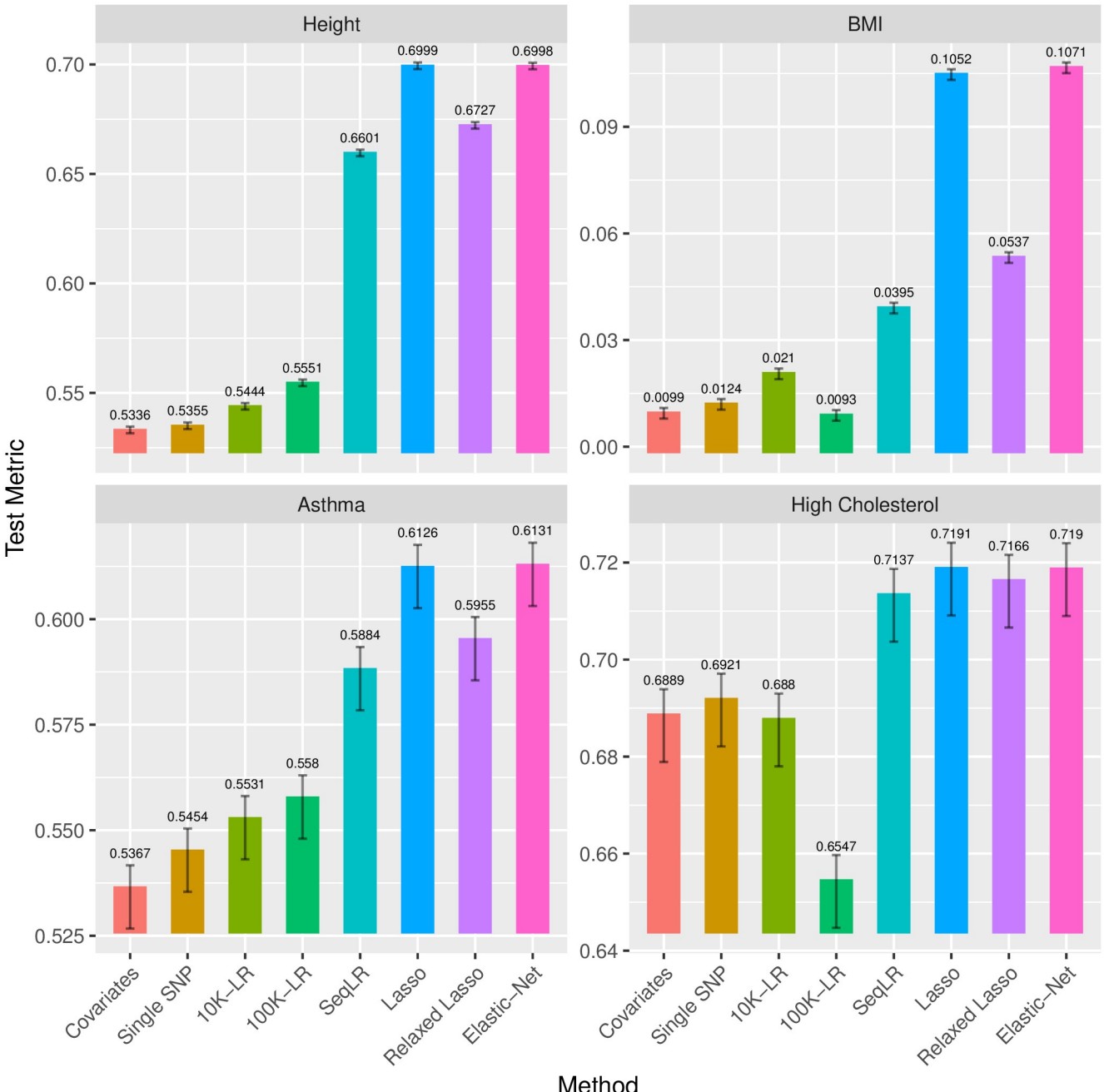

**Fig 2. Comparison of the predictive performance of the different polygenic prediction methods evaluated on the test set.** $R^2$ are evaluated for continuous phenotypes height and BMI, and AUC evaluated for binary phenotypes asthma and high cholesterol. The error bar uses 2 standard errors to show the statistical significance.

`sparse-ldm` subcommand implemented in GCTB version 2.0.1 [29]. Using the GWAS summary statistics computed on the set of individuals and following the GCTB's recommendations, we applied SBayesR with the following options: `gctb --sbayes R --ldm [the LD matrix] --pi 0.95,0.02,0.02,0.01 --gamma 0.0,0.01,0.1,1 --chain-length 10000 --burn-in 4000 --gwas-summary [the GWAS summary statistics]`. We report the model performance on the test set.

We train PRS-CS and SBayesR on the combined training and validation dataset with recommended settings. To make it a fair comparison, for the other methods with tuning parameter(s), we follow a refitting mechanism:

1. fit models on the training set under different parameters;

2. choose the optimal parameter based on the metric ($R^2$/AUC) on the validation set;

3. refit the model with the chosen parameter on a combined training and validation set.

This is often recommended for methods with tuning parameters to make the most of the validation set, and for the lasso/elastic-net, we demonstrate those steps with a code example in the vignette of our **snpnet** package. The predictive performance is compared in Fig 3. SBayesR seems fairly competitive on binary phenotypes, achieving higher test AUC on asthma and high cholesterol. For continuous phenotypes, the lasso and the elastic-net seem to have some advantage, with the lasso doing the best for height and the elastic-net doing the best for BMI. Aside from the predictive performance, SBayesR would always include all the variables in the model, while the lasso/elastic-net class often ends up using only a small fraction of the variables. While prediction is key to the relevance of PRS methods, the sparsity of the solution achieved by the lasso/elastic-net class is also very important for scientific understanding of the genetics behind.

Height is a polygenic and heritable trait that has been studied for a long time. It has been used as a model for other quantitative traits, since it is easy to measure reliably. From twin and sibling studies, the narrow sense heritability is estimated to be 70-80% [30–32]. Recent estimates controlling for shared environmental factors present in twin studies calculate heritability at 0.69 [33, 34]. A linear based model with common SNPs explains 45% of the variance [35] and a model including imputed variants explains 56% of the variance, almost matching the estimated heritability [36]. So far, GWAS studies have discovered 697 associated variants that explain one fifth of the heritability [37, 38]. Recently, a large sample study was able to identify more variants with low frequencies that are associated with height [39]. Using the lasso with the larger UK Biobank dataset allows both a better estimate of the proportion of variance that can be explained by genomic predictors and simultaneous selection of SNPs that may be associated. The results are summarized in Table 1, where for each model class (row), the reported numbers are based on the fitted model on the training set that achieves the best validation performance (if any hyper-parameter). The associated $R^2$ curves for the lasso and the relaxed lasso are shown in Fig 4. The residuals of the optimal lasso prediction are plotted in Fig 5.

A large number (47,673) of SNPs need to be selected in order to achieve the optimal $R^2_{\text{test}} = 0.6999$ for the lasso and similarly for the elastic-net, though it is only a small fraction considering the entire variant set. Comparatively, the relaxed lasso sacrifices some predictive performance by including a much smaller subset of variables (13,395). Past the optimal point, the additional variance introduced by refitting such large models may be larger than the reduction in bias. The large models confirm the extreme polygenicity of standing height.

In comparison to the other models, the lasso performs significantly better in terms of $R^2_{\text{test}}$ than all univariate methods, and outperforms multivariate methods based on univariate $p$-value ordering. That demonstrates the value of simultaneous variable selection and estimation from a multivariate perspective, and enables us to predict height to within 10 cm about 95% of the time based only on SNP information (together with age and sex). We also notice that the sequential linear regression approach does a good job, whose performance gets close to that of the relaxed lasso. It is straightforward and easy to implement using existing softwares such as **PLINK** [13].

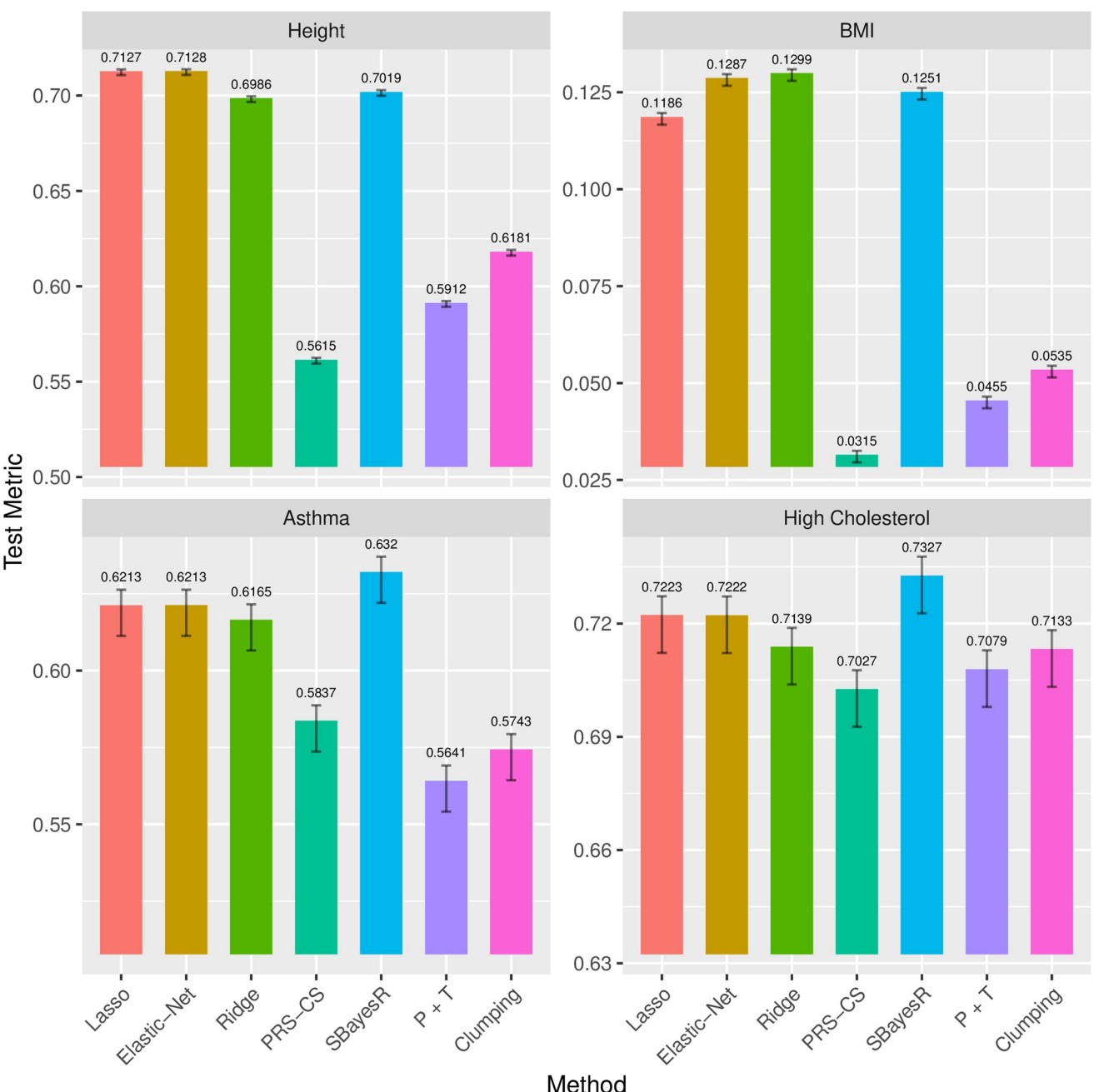

**Fig 3. Comparison of the test set predictive performance of the different polygenic risk score (PRS) methods with refitting on the training and the validation set.** $R^2$ are evaluated for continuous phenotypes height and BMI, and AUC evaluated for binary phenotypes asthma and high cholesterol. The error bar uses 2 standard errors to show the statistical significance.

Recently [21] apply a lasso based method to predict height and other phenotypes on the UK Biobank. Instead of fitting on all QC-satisfied SNPs (as stated in the experiment details paragraph of Materials and methods), they pre-screen 50K or 100K most significant SNPs in terms of the univariate *p*-value and apply lasso on that set only. In addition, although both datasets come from the same UK Biobank, the subset of individuals they used is larger than ours. While we restrict the analysis to the unrelated individuals who have self-reported white British

**Table 1. $R^2$ values for height (without refitting).** For sequential LR, lasso and relaxed lasso, the chosen model is based on maximum $R^2$ on the validation set. Model (3) to (9) each includes Model (2) plus their own specification as stated in the Form column. The elastic-net picks $\alpha = 0.9$ based on the validation performance.

| Model | Form | $R^2_{\text{train}}$ | $R^2_{\text{val}}$ | $R^2_{\text{test}}$ | Size |
|-------|------|------|------|------|------|
| (1) | Age + Sex | 0.5300 | 0.5260 | 0.5288 | 2 |
| (2) | Age + Sex + 10 PCs | 0.5344 | 0.5304 | 0.5336 | 12 |
| (3) | Strong Single SNP | 0.5364 | 0.5323 | 0.5355 | 13 |
| (4) | 10K Combined | 0.5482 | 0.5408 | 0.5444 | 10,012 |
| (5) | 100K Combined | 0.5833 | 0.5515 | 0.5551 | 100,012 |
| (6) | Sequential LR | 0.7416 | 0.6596 | 0.6601 | 17,012 |
| (7) | Lasso | 0.8304 | 0.6992 | **0.6999** | 47,673 |
| (8) | Relaxed Lasso | 0.7789 | 0.6718 | 0.6727 | 13,395 |
| (9) | Elastic Net | 0.8282 | 0.6991 | 0.6998 | 48,268 |

ancestry, they look at Europeans including British, Irish and Any Other White. For a fair comparison, we follow their procedure (pre-screening 100K SNPs) but run on our subset of the dataset. The results are shown in Table 2. We see that the improvement of the full lasso over the prescreened lasso is almost 0.5% in test $R^2$, and 1% relative to the proportion of residual variance explained after covariate adjustment.

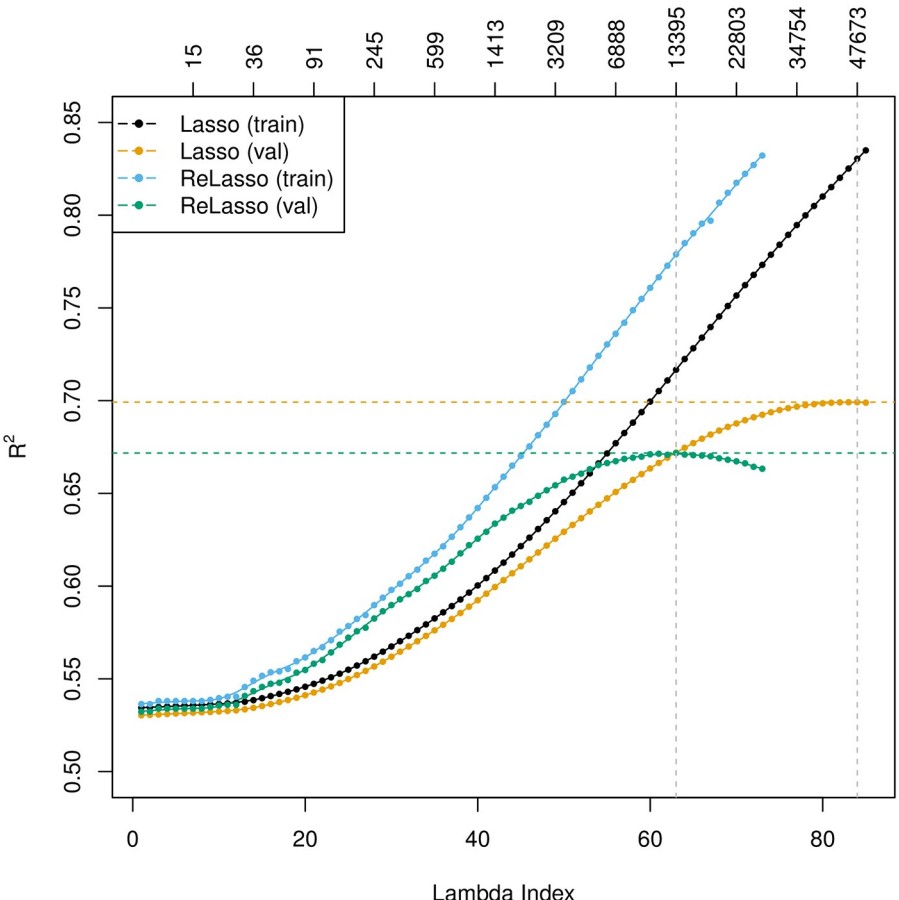

**Fig 4. $R^2$ plot for height.** The primary horizontal axis on the bottom represents the index of lambda values, $1 \leq \ell \leq L$, which correspond to the sequence of the regularization parameters, $\lambda_1 > \lambda_2 > \cdots > \lambda_L$. The top axis shows the number of active variables in the model. ReLasso: relaxed lasso.

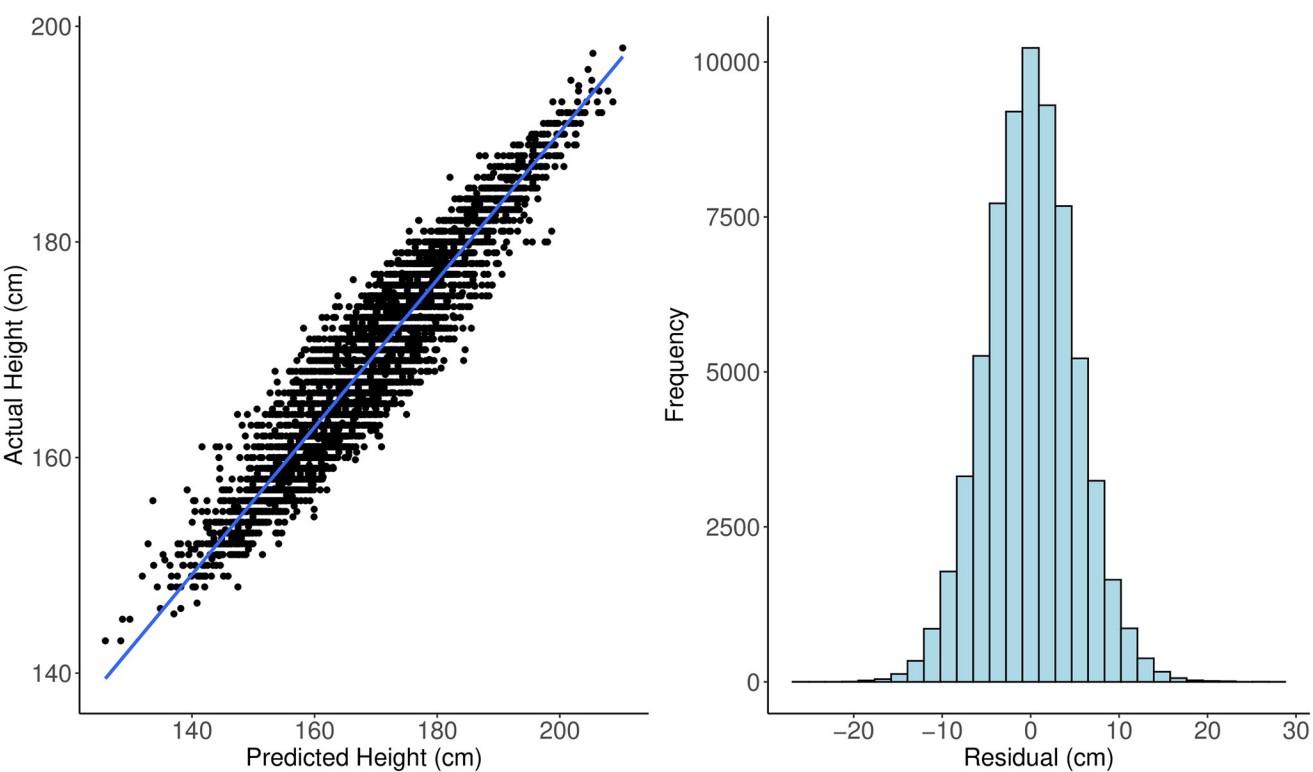

**Fig 5.** Left: actual height versus predicted height on 5000 random samples from the test set. A regression line with its 95% confidence band is also added on top of the dots. The correlation between actual height and predicted height is 0.9416. Right: histogram of the lasso residuals for height. Standard deviation of the residual is 5.05 (cm).

In addition, we compare the full lasso coefficients and the univariate $p$-values from GWAS in Fig 6. The vertical grey dotted line indicates the top 100K cutoff in terms of $p$-value. We see although a general decreasing trend appears in the magnitude of the lasso coefficients with respect to increasing $p$-values (decreasing $-\log_{10}(p)$), there are a number of spikes even in the large $p$-value region which is considered marginally insignificant. This shows that variants beyond the strongest univariate ones contribute to prediction.

We conduct the lasso and elastic-net with the refitting mechanism and compare them with the other well-established PRS methods. From Table 3, we see that the lasso and the elastic-net do the best job and also uses only a small fraction of the variables.

## Discussion

In this paper, we propose a novel batch screening iterative lasso (BASIL) algorithm to fit the full lasso solution path for very large and high-dimensional datasets. It can be used, among the

**Table 2. Comparison of prediction results on height with the model trained following the same procedure as ours except for an additional prescreening step as done in [21].** In addition to $R^2$, proportion of residual variance explained (denoted by $h^2_{test}$) and correlation between the fitted values and actual values are computed. We also compute an adjusted correlation between the residual after regressing age and sex out from the prediction and the residual after regressing age and sex out from the true response, both on the test set.

| Method | $R^2_{val}$ | $R^2_{test}$ | $h^2_{test}$ | $\mathbf{Cor_{test}}$ | $\mathbf{Cor_{test}}$−{age, sex} |
|---|---|---|---|---|---|
| Lasso | 69.92% | 69.99% | 35.66% | 0.8366 | 0.4079 |
| Prescreened lasso | 69.40% | 69.56% | 34.73% | 0.8340 | 0.4025 |

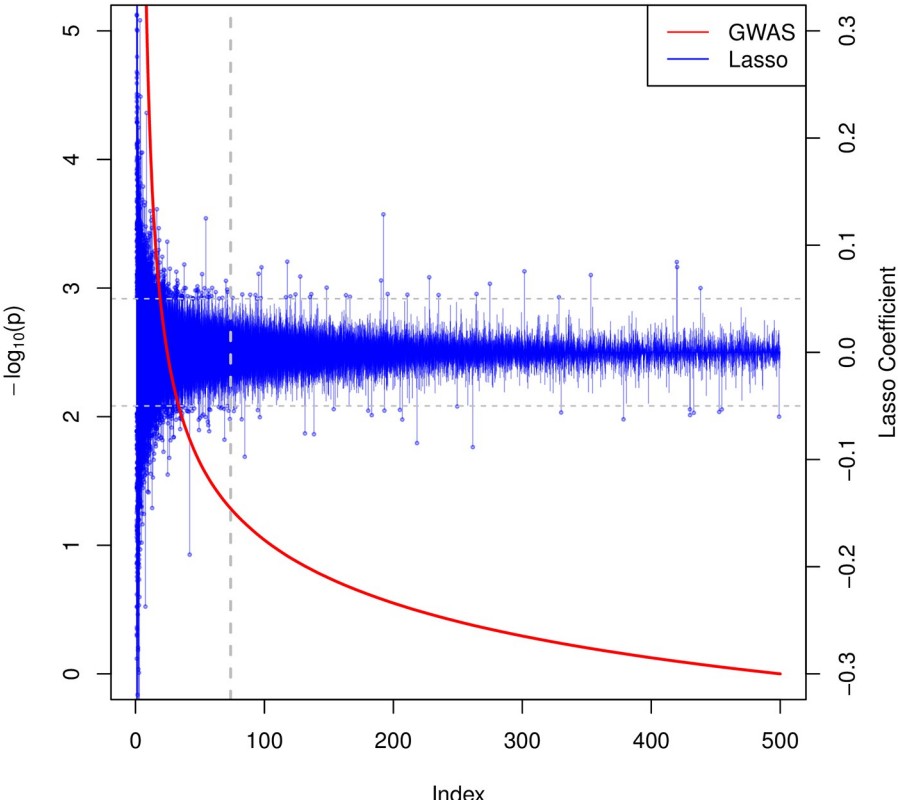

**Fig 6. Comparison of the lasso coefficients and univariate *p*-values for height.** The index on the horizontal axis represents the SNPs sorted by their univariate *p*-values. The red curve associated with the left vertical axis shows the $-\log_{10}$ of the univariate *p*-values. The blue bars associated with the right vertical axis show the corresponding lasso coefficients for each (sorted) SNP. The horizontal dotted lines in gray identifies lasso coefficients of ±0.05. The vertical one represents the 100K cutoff used in [21].

others, for Gaussian linear model, logistic regression and Cox regression, and can be easily extended to fit the elastic-net with mixed $\ell_1/\ell_2$ penalty. It enjoys the advantages of high efficiency, flexibility and easy implementation. For SNP data as in our applications, we develop an R package **snpnet** that incorporates SNP-specific optimizations and are able to process datasets of wide interest from the UK Biobank.

In our algorithm, the choice of $M$ is important for the practical performance. It trades off between the number of iterations and the computation per iteration. With a small $M$ or small update of the strong set, it is very likely that we are unable to proceed fast along the $\lambda$ sequence

**Table 3. $R^2$ values for height by different PRS methods with refitting.**

| Model | Form | $R^2_{\text{test}}$ | Size |
|---|---|---|---|
| (1) | Lasso | 0.7127 | 45,653 |
| (2) | Elastic Net | **0.7128** | 45,549 |
| (3) | Ridge | 0.6986 | 175,012 |
| (4) | PRS-CS | 0.5615 | 148,064 |
| (5) | SBayesR | 0.7019 | 667,057 |
| (6) | P + T | 0.5912 | 15,544 |
| (7) | Clumping | 0.6181 | 17,433 |

in each iteration. Although the design of the BASIL algorithm guarantees that for any $M$, $\Delta M > 0$, we are able to obtain the full solution path after sufficient iterations, many iterations will be needed if $M$ is chosen too small, and the disk I/O cost will be dominant. In contrast, a large $M$ will incur more memory burden and more expensive lasso computation, but with the hope to find more valid lasso solutions in one iteration, save the number of iterations and the disk I/O. It is hard to identify the optimal $M$ a priori. It depends on the computing architecture, the size of the problem, the nature of the phenotype, etc. For this reason, we tend to leave it as a subjective parameter to the user's choice. However in the meantime, we do plan to provide a more systematic option to determine $M$, which leverages the strong rules again. Recall that in the simple setting with no intercept and no covariates, the initial strong set is constructed by $|x_j^\top y| \leq 2\lambda - \lambda_{\max}$. Since the strong rules rarely make mistakes and are fairly effective in discarding inactive variables, we can guide the choice of batch size $M$ by the number of $\lambda$ values we want to cover in the first iteration. For example, one may want the strong set to be large enough to solve for the first 10 $\lambda$'s in the first iteration. We can then let $M = |\{1 \leq j \leq p : |x_j^\top y| > 2\lambda_{10} - \lambda_{\max}\}|$. Despite being adaptive to the data in some sense, this approach is by no means computationally optimal. It is more based on heuristics that the iteration should make reasonable progress along the path.

Our numerical studies demonstrate that the iterative procedure effectively reduces a big-$n$-big-$p$ lasso problem into one that is manageable by in-memory computation. In each iteration, we are able to use parallel computing when applying screening rules to filter out a large number of variables. After screening, we are left with only a small subset of data on which we are able to conduct intensive computation like cyclical coordinate descent all in memory. For the subproblem, we can use existing fast procedures for small or moderate-size lasso problems. Thus, our method allows easy reuse of previous software with lightweight development effort.

When a large number of variables is needed in the optimal predictive model, it may still require either large memory or long computation time to solve the smaller subproblem. In that case, we may consider more scalable and parallelizable methods like proximal gradient descent [40] or dual averaging [41, 42]. One may think why don't we directly use these methods for the original full problem? First, the ultra high dimension makes the evaluation of gradients, even on mini-batch very expensive. Second, it can take a lot more steps for such first-order methods to converge to a good objective value. Moreover, the speed of convergence depends on the choice of other parameters such as step size and additional constants in dual averaging. For those reasons, we still prefer the tuning-free and fast coordinate descent methods when the subproblem is manageable.

The lasso has nice variable selection and prediction properties if the linear model assumption together with some additional assumptions such as the restricted eigenvalue condition [43] or the irrepresentable condition [44] holds. In practice, such assumptions do not always hold and are often hard to verify. In our UK Biobank application, we don't attempt to verify the exact conditions, and the selected model can be subject to false positives. However, we demonstrate relevance of the selection via empirical consistency with the GWAS results. We have seen superior prediction performance by the lasso as a regularized regression method compared to other methods. More importantly, by leveraging the sparsity property of the lasso, we are able to manage the ultrahigh-dimensional problem and obtain a computationally efficient solution.

When comparing with other methods in the UK Biobank experiments, due to the large number of test samples (60,000+), we are confident that the lasso and the elastic-net methods are able to do significantly better than all other methods on height and BMI, and are as competitive as SBayesR on asthma and high cholesterol. In fact, the standard error of $R^2$ can be easily derived by the delta method, and the standard error of the AUC can be estimated and

upper bounded by $1/(4 \min(m, n))$ [45, 46], where $m, n$ represents the number of positive and negative samples. For height and BMI, it turns out that the standard errors are roughly 0.001, or 0.1%. For asthma and high cholesterol, considering the case rate around 12%, the standard errors can be upper bounded by 0.005, or 0.5%. The estimated standard errors are reflected in the error bars in Figs 2 and 3. Therefore, speaking of the predictive performance, on height and BMI, the lasso/elastic-net class performs significantly better than the other methods, while on asthma and high cholesterol, the lasso/elastic-net and the SBayesR are both fairly competitive—their difference is not statistically significant. Moreover, the lasso/elastic-net method builds parsimonious models using only a small fraction of the variants. It is more interpretable and can have meaningful implications on the genetics behind.

## Materials and methods

### Variants in the BASIL framework

Some other very useful components can be easily incorporated into the BASIL framework. We will discuss debiasing using the relaxed lasso and the inclusion of adjustment covariates.

The lasso is known to shrink coefficients to exclude noise variables, but sometimes such shrinkage can degrade the predictive performance due to its effect on actual signal variables. [24] introduces the relaxed lasso to correct for the potential over-shrinkage of the original lasso estimator. They propose a refitting step on the active set of the lasso solution with less regularization, while a common way of using it is to fit a standard OLS on the active set. The active set coefficients are then set to

$$\hat{\beta}_{\mathcal{A},\text{Relax}}(\lambda) = \underset{\beta_{\mathcal{A}} \in \mathbb{R}^{|\mathcal{A}|}}{\operatorname{argmin}} \|y - X_{\mathcal{A}}\beta_{\mathcal{A}}\|_2^2,$$

whereas the coefficients for the inactive set remain at 0. This refitting step can revert some of the shrinkage bias introduced by the vanilla lasso. It doesn't always reduce prediction error due to the accompanied increase in variance when there are many variables in the model or when the signals are weak. That being said, we can still insert a relaxed lasso step with little effort in our iterative procedure: once a valid lasso solution is found for a new $\lambda$, we may refit with OLS. As we iterate, we can monitor validation error for the lasso and the relaxed lasso. The relaxed lasso will generally end up choosing a smaller set of variables than the lasso solution in the optimal model.

In some applications such as GWAS, there may be confounding variables $Z \in \mathbb{R}^{n \times q}$ that we want to adjust for in the model. Population stratification, defined as the existence of a systematic ancestry difference in the sample data, is one of the common factors in GWAS that can lead to spurious discoveries. This can be controlled for by including some leading principal components of the SNP matrix as variables in the regression [47]. In the presence of such variables, we instead solve

$$(\hat{\alpha}(\lambda), \hat{\beta}(\lambda)) = \underset{\alpha \in \mathbb{R}^q, \beta \in \mathbb{R}^p}{\operatorname{argmin}} \ \frac{1}{2n}\|y - Z\alpha - X\beta\|_2^2 + \lambda\|\beta\|_1. \tag{7}$$

This variation can be easily handled with small changes in the algorithm. Instead of initializing the residual with the response $y$, we set $r^{(0)}$ equal to the residual from the regression of $y$ on the covariates. In the fitting step, in addition to the variables in the strong set, we include the covariates but leave their coefficients unpenalized as in (7). Notice that if we want to find relaxed lasso fit with the presence of adjustment covariates, we need to include those covariates

in the OLS as well, i.e.,

$$(\hat{\alpha}_{\text{Relax}}(\lambda), \hat{\beta}_{\mathcal{A},\text{Relax}}(\lambda)) = \underset{\alpha \in \mathbb{R}^q, \beta_{\mathcal{A}} \in \mathbb{R}^{|\mathcal{A}|}}{\text{argmin}} \|y - Z\alpha - X_{\mathcal{A}}\beta_{\mathcal{A}}\|_2^2. \tag{8}$$

## UK Biobank experiment details

We focused on 337,199 White British unrelated individuals out of the full set of over 500,000 from the UK Biobank dataset [11] that satisfy the same set of population stratification criteria as in [22]: (1) self-reported White British ancestry, (2) used to compute principal components, (3) not marked as outliers for heterozygosity and missing rates, (4) do not show putative sex chromosome aneuploidy, and (5) have at most 10 putative third-degree relatives. These criteria are meant to reduce the effect of confoundedness and unreliable observations.

The number of samples is large in the UK Biobank dataset, so we can afford to set aside an independent validation set without resorting to the costly cross-validation to find an optimal regularization parameter. We also leave out a subset of observations as test set to evaluate the final model. In particular, we randomly partition the original dataset so that 60% is used for training, 20% for validation and 20% for test. The lasso solution path is fit on the training set, whereas the desired regularization is selected on the validation set, and the resulting model is evaluated on the test set.

We are going to further discuss some details in our application that one might also encounter in practice. They include adjustment for confounders, missing value imputation and variable standardization in the algorithm.

In genetic studies, spurious associations are often found due to confounding factors. Among the others, one major source is the so-called population stratification [48]. To adjust for that effect, it is common is to introduce the top principal components and include them in the regression model. Therefore in the lasso method, we are going to solve (7) where in addition to the SNP matrix $X$, we let $Z$ include covariates such as age, sex and the top 10 PCs of the SNP matrix.

Missing values are present in the dataset. As quality control normally done in genetics, we first discard observations whose phenotypic value of interest is not available. We further exclude variants whose missing rate is greater than 10% or the minor allele frequency (MAF) is less than 0.1%, which results in around 685,000 SNPs for height. In particulr, 685,362 for height, 685,371 for BMI, 685,357 for asthma and 685,357 for HC. The number varies because the criteria are evaluated on the subset of individuals whose phenotypic value is observed (after excluding the missing ones), which can be different across different phenotypes. For those remaining variants, mean imputation is conducted to fill the missing SNP values; that is, the missing values in every SNP are imputed with the mean observed level of that SNP in the population under study.

When it comes to the lasso fitting, there are some subtleties that can affect its variable selection and prediction performance. One of them is variable standardization. It is often a step done without much thought to deal with heterogeneity in variables so that they are treated fairly in the objective. However in our studies, standardization may create some undesired effect. To see this, notice that all the SNPs can only take values in 0, 1, 2 and NA—they are already on the same scale by nature. As we know, standardization would use the current standard deviation of each predictor as the divisor to equalize the variance across all predictors in the lasso fitting that follows. In this case, standardization would unintentionally inflate the magnitude of rare variants and give them an advantage in the selection process since their coefficients effectively receive less penalty after standardization. In Fig 7, we can see the

## Histogram of SNP Standard Deviation

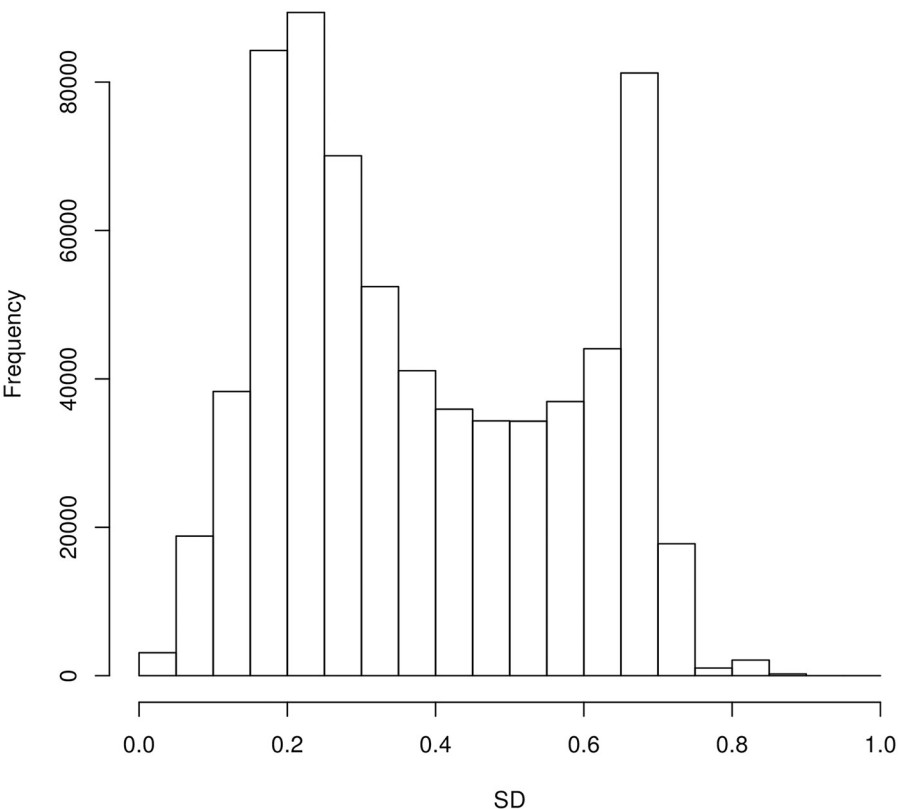

**Fig 7. Histogram of the standard deviations of the SNPs.** They are computed *after* mean imputation of the missing values because they would be the exact standardization factors to be used if the lasso were applied with variable standardization on the mean-imputed SNP matrix.

distribution of standard deviation across all variants in our dataset. Hence, to avoid potential spurious findings, we choose not to standardize the variants in the experiments.

## Computational optimization in software implementation

Among the iterative steps in BASIL, screening and checking are where we need to deal with the full dataset. To deal with the memory bound, we can use memory-mapped I/O. In R, **big-memory** [49] provides a convenient implementation for that purpose. That being said, we do not want to rely on that for intensive computation modules such as cyclic coordinate descent, because frequent visits to the on-disk data would still be slow. Instead, since the subset of strong variables would be small, we can afford to bring them to memory and do fast lasso fitting there. We only use the full memory-mapped dataset in KKT checking and screening. Moreover since checking in the current iteration can be done together with the screening in the next iteration, effectively only one expensive pass over the full dataset is needed every iteration.

In addition, we use a set of techniques to speed up the computation. First, the KKT check can be easily parallelized by splitting on the features when multi-core machines are available. The speedup of this part is immediate and (slightly less than) proportional to the number of

**Table 4. Timing performance (milliseconds) on multiplication of SNP matrix and residual matrix.** The methods are all implemented in C++ and run on a Macbook with 2.9 GHz Intel Core i7 and 8 GB 1600 MHz DDR3.

| Multiplication Method | $n = 200, p = 800$ | $n = 2000, p = 8000$ |
|---|---|---|
| Standard | 3.20 | 306.01 |
| SNP-Optimized | 1.32 | 130.21 |

cores available. Second, specific to the application, we exploit the fact that there are only 4 levels for each SNP value and design a faster inner product routine to replace normal float number multiplication in the KKT check step. In fact, given any SNP vector $x \in \{0, 1, 2, \mu\}^n$ where $\mu$ is the imputed value for the missing ones, we can write the dot product with a vector $r \in \mathbb{R}^n$ as

$$x^\top r = \sum_{i=1}^n x_i r_i = 1 \cdot \sum_{i:x_i=1} r_i + 2 \cdot \sum_{i:x_i=2} r_i + \mu \cdot \sum_{i:x_i=\mu} r_i. \tag{9}$$

We see that the terms corresponding to 0 SNP value can be ignored because they don't contribute to the final result. This will significantly reduce the number of arithmetic operations needed to compute the inner product with rare variants. Further, we only need to set up 3 registers, each for one SNP value accumulating the corresponding terms in $r$. A series of multiplications is then converted to summations. In our UK Biobank studies, although the SNP matrix is not sparse enough to exploit sparse matrix representation, it still has around 70% 0's. We conduct a small experiment to compare the time needed to compute $X^\top R$, where $X \in \{0, 1, 2, 3\}^{n \times p}, R \in \mathbb{R}^{p \times k}$. The proportions for the levels in $X$ are about 70%, 10%, 10%, 10%, similar to the distribution of SNP levels in our study, and $R$ resembles the residual matrix when checking the KKT condition. The number of residual vectors is $k = 20$. The mean time over 100 repetitions is shown in Table 4.

We implement the procedure with all the optimizations above in an R package called **snpnet**, which is currently available at https://github.com/junyangq/snpnet and will be submitted to the CRAN repository of R packages. While most of the numerical experiments throughout the paper are based on an earlier version of the package (available in the V1.0 branch of the repository) assuming bed file format provided by **PLINK** 1.9, we highly recommend one to use the current version that works with the pgen file format provided by **PLINK** 2.0 [13, 28]. It takes advantage of **PLINK** 2.0's new R interface as well as its efficient --variant-score module for matrix multiplication. The module exploits combined techniques of multithreading, a good linear algebra library, and an alternate code path for very-low-MAF SNPs (similar to the one proposed in (9)) to make the computation even faster. In order to achieve better efficiency in each lasso fitting, we suggest using **snpnet** together with **glmnetPlus**, a warm-started version of **glmnet**, which is currently available at https://github.com/junyangq/glmnetPlus. It allows one to provide a good initialization of the coefficients to fit part of the solution path instead of always starting from the all-zero solution by **glmnet**.

## Related methods and packages

There are a number of existing screening rules for solving big lasso problems. [50] use a screened set to scale down the logistic lasso problem and check the KKT condition to validate the solution. Their focus, however, is on selecting a lasso model of particular size and only the initial screened set is expanded if the KKT condition is violated. In contrast, we are interested in finding the whole solution path (before overfitting). We adopt a sequential approach and

keep updating the screened set at each iteration. This allows us to potentially keep the screened set small as we move along the solution path. Other rules include the SAFE rule [51], Sure Independence Screening [52], and the DPP and EDPP rules [53].

We expand the discussion on these screening rules a bit. [52] exploits marginal information of correlation to conduct screening but the focus there is not optimization algorithm. Most of the screening rules mentioned above (except for EDPP) use inner product with the current residual vector to measure the importance of each predictor at the next $\lambda$—those under a threshold can be ignored. The key difference across those rules is the threshold defined and whether the resulting discard is safe. If it is safe, one can guarantee that only one iteration is needed for each $\lambda$ value, compared with others that would need more rounds if an active variable was falsely discarded. Though the strong rules rarely make this mistake, safe screening is still a nice feature to have in single-$\lambda$ solutions. However, under the batch mode we consider due to the desire of reducing the number of full passes over the dataset, the advantage of safe threshold may not be as much. In fact, one way we might be able to leverage the safe rules in the batch mode is to first find out the set of candidate predictors for the several $\lambda$ values up to $\lambda_k$ we wish to solve in the next iteration based on the current inner products and the rules' safe threshold, and then solve the lasso for these parameters. Since these rules can often be conservative, we would then have strong incentive to solve for, say, one further $\lambda$ value $\lambda_{k+1}$ because if the current screening turns out to be a valid one as well, we will find one more lasso solution and move one step forward along the $\lambda$ sequence we want to solve for. This can potentially save one iteration of the procedure and thus one expensive pass over the dataset. The only cost there is computing the lasso solution for one more $\lambda_{k+1}$ and computing inner products with one more residual vector at $\lambda_{k+1}$ (to check the KKT condition). The latter can be done in the same pass as we compute inner products at $\lambda_k$ for preparing the screening in the next iteration, and so no additional pass is needed. Thus under the batch mode, the property of safe screening may not be as important due to the incentive of aggressive model fitting. Nevertheless it would be interesting to see in the future EDPP-type batch screening. It uses inner products with a modification of the residual vector. Our algorithm still focuses of inner products with the vanilla residual vector.

To address the large-scale lasso problems, several packages have been developed such as **biglasso** [54], **bigstatsr** [55], **oem** [56] and the lasso routine from **PLINK** 1.9 [13, 14].

Among them, **oem** specializes in tall data (big $n$) and can be slow when $p > n$. In many real data applications including ours, the data are both large-sample and high-dimensional. However, we might still be able to use **oem** for the small lasso subroutine since a large number of variables have already been excluded. The other packages, **biglasso**, **bigstatsr**, **PLINK** 1.9, all provide efficient implementations of the pathwise coordinate descent with warm start. **PLINK** 1.9 is specifically developed for genetic datasets and is widely used in GWAS and research in population genetics. In **bigstatsr**, the `big_spLinReg` function adapts from the `biglasso` function in **biglasso** and incorporates a Cross-Model Selection and Averaging (CMSA) procedure, which is a variant of cross-validation that saves computation by directly averaging the results from different folds instead of retraining the model at the chosen optimal parameter. They both use memory-mapping to process larger-than-RAM, on-disk datasets as if they were in memory, and based on that implement coordinate descent with strong rules and warm start.

The main difference between BASIL and the algorithm these packages use is that BASIL tries to solve a series of models every full scan of the dataset (at checking and screening) and thus effectively reduce the number of passes over the dataset. This difference may not be significant in small or moderate-sized problems, but can be critical in big data applications especially when the dataset cannot be fully loaded into the memory. A full scan of a larger-than-

RAM dataset can incur a lot of swap-in/out between the memory and the disk, and thus a lot of disk I/O operations, which is known to be orders of magnitude slower than in-memory operations. Thus reducing the number of full scans can greatly improve the overall performance of the algorithm.

Aside from potential efficiency consideration, all of those packages aforementioned have to re-implement a variety of features existent in many small-data solutions but for big-data context. Nevertheless, currently they don't provide as much functionality as needed in our real-data application. First, in the current implementations, **PLINK** 1.9 only supports the Gaussian family, **biglasso** and **bigstatsr** only supports the Gaussian and binomial families, whereas **snpnet** can easily extend to other regression families and already built in Gaussian, binomial and Cox families. Also, **biglasso**, **bigstatsr** and **PLINK** 1.9 all standardize the predictors beforehand, but in many applications such as our UK Biobank studies, it is more reasonable to leave the predictors unstandardized. In addition, it can take some effort to convert the data to the desired format by these packages. This would be a headache if the raw data is in some special format and one cannot afford to first convert the full dataset into an intermediate format for which a tool is provided to convert to the desired one by **biglasso** or **bigstatsr**. This can happen, for example, if the raw data is highly compressed in a special format. For the BED binary format we work with in our application, `readRAW_big.matrix` function from **BGData** can convert a raw file to a `big.matrix` object desired by **biglasso**, and `snp_readBed` function from **bigsnpr** [55] allows one to convert it to `FBM` object desired by **bigstatsr**. However, **bigsnpr** doesn't take input data that has any missing values, which can prevalent in an SNP matrix (as in our application). Although **PLINK** 1.9 works directly with the BED binary file, its lasso solver currently only supports the Gaussian family, and it doesn't return the full solution path. Instead it returns the solution at the smallest $\lambda$ value computed and needs a good heritability estimate as input from the user, which may not be immediately available.

We summarize the main advantages of the BASIL algorithm:

- **Input data flexibility**. Our algorithm allows one to deal directly with any data type as long as the screening and checking steps are implemented, which is often very lightweight development work like matrix multiplication. This can be important in large-scale applications especially when the data is stored in a compressed format or a distributed way since then we would not need to unpack the full data and can conduct KKT check and screening on its original format. Instead only a small screened subset of the data needs to be converted to the desired format by the lasso solver in the fitting step.

- **Model flexibility**. We can easily transfer the modeling flexibility provided by existing packages to the big data context, such as the options of standardization, sample weights, lower/upper coefficient limits and other families in generalized linear models provided by existing packages such as **glmnet**. This can be useful, for example, when we may not want to standardize predictors already in the same unit to avoid unintentionally different penalization of the predictors due to difference in their variance.

- **Effortless development**. The BASIL algorithm allows one to maximally reuse the existing lasso solutions for small or moderate-sized problems. The main extra work would be an implementation of batch screening and KKT check with respect to a particular data type. For example, in the **snpnet** package, we are able to quickly extend the in-memory **glmnet** solution to large-scale, ultrahigh-dimensional SNP data. Moreover, the existing convenient data interface provided by the **BEDMatrix** package further facilitates our implementation.

**Table 5. Timing comparison on a synthetic dataset of size $n = 50,000$ and $p = 100,000$.** The time for bigstatsr and biglasso has two components: one for the conversion to the desired data type and the other for the actual computation. The experiments are all run with 16 cores and 64 GB memory.

| R Package | Elapsed Time (minutes) |
|---|---|
| **bigstatsr** [55] | 2.93 + 56.80 |
| **bigstatsr** + CMSA [55] | 2.93 + 101.75 |
| **biglasso** [54] | 4.55 + 54.27 |
| **PLINK** [13, 14] | 53.52 |
| **snpnet** | **44.79** |

- **Computational efficiency**. Our design reduces the number of visits to the original data that sits on the disk, which is crucial to the overall efficiency as disk read can be orders of magnitude slower than reading from the RAM. The key to achieving this is to bring batches of promising variables into the main memory, hoping to find the lasso solutions for more than one $\lambda$ value each iteration and check the KKT condition for those $\lambda$ values in one pass of the entire dataset.

Lastly, we are going to provide some timing comparison with existing packages. As mentioned in previous sections, those packages provide different functionalities and have different restrictions on the dataset. For example, most of them (**biglasso**, **bigstatsr**) assume that there are no missing values, or the missing ones have already been imputed. In **bigsnpr**, for example, we shouldn't have SNPs with 0 MAF either. Some packages always standardize the variants before fitting the lasso. To provide a common playground, we create a synthetic dataset with no missing values, and follow a standardized lasso procedure in the fitting stage, simply to test the computation. The dataset has 50,000 samples and 100,000 variables, and each takes value in the SNP range, i.e., in 0, 1, or 2. We fit the first 50 lasso solutions along a prefix $\lambda$ sequence that contains 100 initial $\lambda$ values (like early stopping for most phenotypes). The total time spent is displayed in Table 5. For **bigstatsr**, we include two versions since it does cross-validation by default. In one version, we make it comply with our single train/val/test split, while in the other version, we use its default 10-fold cross-validation version—Cross-Model Selection and Averaging (CMSA). Notice that the final solution of iCMSA is different from the exact lasso solution on the full data because the returned coefficient vector is a linear combination of the coefficient vectors from the 10 folds rather than from a retrained model on the full data. We uses 128GB memory and 16 cores for the computation.

From the table, we see that **snpnet** is at about 20% faster than other packages concerned. The numbers before the "+" sign are the time spent on converting the raw data to the required data format by those packages. The second numbers are time spent on actual computation.

It is important to note though that the performance relies not only on the algorithm, but also heavily on the implementations. The other packages in comparison all have their major computation done with C++ or Fortran. Ours, for the purpose of meta algorithm where users can easily integrate with any lasso solver in R, still has a significant portion (the iterations) in R and multiple rounds of cross-language communication. That can degrade the timing performance to some degree. If there is further pursuit of speed performance, there is still space for improvement by more designated implementation.

## Supporting information

**S1 Appendix. Results for body mass index (BMI).**
(PDF)

**S2 Appendix. Results for asthma.**
(PDF)

**S3 Appendix. Results for high cholesterol.**
(PDF)

**S4 Appendix. Manhattan plots.**
(PDF)

**S1 Table. $R^2$ values for BMI (without refitting).** $R^2$ values for BMI (without refitting). For lasso and relaxed lasso, the chosen model is based on maximum $R^2$ on the validation set. Model (3) to (9) each includes Model (2) plus their own specification as stated in the Form column. The elastic-net picks $\alpha = 0.1$ based on the validation performance.
(PDF)

**S2 Table. $R^2$ values for body mass index by different PRS methods with refitting.**
(PDF)

**S3 Table. AUC values for asthma (without refitting).** For lasso and relaxed lasso, the chosen model is based on maximum AUC on the validation set. Model (3) to (9) each includes Model (2) plus their own specification as stated in the Form column. The elastic-net picks $\alpha = 0.1$ based on the validation performance.
(PDF)

**S4 Table. AUC values for asthma by different PRS methods with refitting.**
(PDF)

**S5 Table. AUC values for high cholesterol (without refitting).** For lasso and relaxed lasso, the chosen model is based on maximum AUC on the validation set. Model (3) to (9) each includes Model (2) plus their own specification as stated in the Form column. The elastic-net picks $\alpha = 0.9$ based on the validation performance.
(PDF)

**S6 Table. AUC values for high cholesterol by different PRS methods with refitting.**
(PDF)

**S1 Fig. $R^2$ plot for body mass index.** The primary horizontal axis on the bottom represents the index of lambda values, $1 \leq \ell \leq L$, which correspond to the sequence of the regularization parameters, $\lambda_1 > \lambda_2 > \cdots > \lambda_L$. The top axis shows the number of active variables in the model. ReLasso: relaxed lasso.
(TIF)

**S2 Fig. Actual body mass index (BMI) versus predicted BMI on 5000 random samples from the test set.** A regression line with its 95% confidence band is also added on top of the dots. The correlation between actual BMI and predicted BMI is 0.3256.
(TIF)

**S3 Fig. Residuals of lasso prediction for body mass index.** Standard deviation of the residual is 4.51 kg/m$^2$.
(TIF)

**S4 Fig. AUC plot for asthma.** The primary horizontal axis on the bottom represents the index of lambda values, $1 \leq \ell \leq L$, which correspond to the sequence of the regularization parameters, $\lambda_1 > \lambda_2 > \cdots > \lambda_L$. The top axis shows the number of active variables in the model.

ReLasso: relaxed lasso.
(TIF)

**S5 Fig. Box plot of the percentile of the linear prediction score among cases versus controls for asthma.** This is based on the optimal lasso model.
(TIF)

**S6 Fig. Stratified prevalence across different percentile bins based on the predicted scores for asthma.** This is based on the optimal lasso model.
(TIF)

**S7 Fig. AUC plot for high cholesterol.** The primary horizontal axis on the bottom represents the index of lambda values, $1 \leq \ell \leq L$, which correspond to the sequence of the regularization parameters, $\lambda_1 > \lambda_2 > \cdots > \lambda_L$. The top axis shows the number of active variables in the model. ReLasso: relaxed lasso.
(TIF)

**S8 Fig. Box plot of the percentile of the linear prediction score among cases versus controls for high cholesterol.** This is based on the optimal lasso model.
(TIF)

**S9 Fig. Stratified prevalence across different percentile bins based on the predicted scores for high cholesterol.** This is based on the optimal lasso model.
(TIF)

**S10 Fig. ROC curve for asthma.** This is based on the optimal lasso model.
(TIF)

**S11 Fig. ROC curve for high cholesterol.** This is based on the optimal lasso model.
(TIF)

**S12 Fig. Manhattan plot of the univariate *p*-values for height.** This is based on the optimal lasso model. The vertical axis shows $-\log_{10}(p)$ for each SNP. The red horizontal line represents a reference level of $p = 5 \times 10^{-8}$.
(TIF)

**S13 Fig. Manhattan plot of the univariate *p*-values for body mass index.** This is based on the optimal lasso model. The vertical axis shows $-\log_{10}(p)$ for each SNP. The red horizontal line represents a reference level of $p = 5 \times 10^{-8}$.
(TIF)

**S14 Fig. The lasso coefficients for height.** This is based on the optimal lasso model. The vertical axis shows the magnitude of the coefficients from **snpnet**. The SNPs with relatively large lasso coefficients are highlighted in green.
(TIF)

**S15 Fig. The lasso coefficients for body mass index.** This is based on the optimal lasso model. The vertical axis shows the magnitude of the coefficients from **snpnet**. The SNPs with relatively large lasso coefficients are highlighted in green.
(TIF)

**S16 Fig. Manhattan plot of the univariate *p*-values for asthma.** This is based on the optimal lasso model. The vertical axis shows $-\log_{10}(p)$ for each SNP. The red horizontal line represents a reference level of $p = 5 \times 10^{-8}$.
(TIF)

**S17 Fig. Manhattan plot of the univariate *p*-values for high cholesterol.** This is based on the optimal lasso model. The vertical axis shows $-\log_{10}(p)$ for each SNP. The red horizontal line represents a reference level of $p = 5 \times 10^{-8}$.
(TIF)

**S18 Fig. The lasso coefficients for asthma.** This is based on the optimal lasso model. The vertical axis shows the magnitude of the coefficients from **snpnet**. The SNPs with relatively large lasso coefficients are highlighted in green.
(TIF)

**S19 Fig. The lasso coefficients for high cholesterol.** This is based on the optimal lasso model. The vertical axis shows the magnitude of the coefficients from **snpnet**. The SNPs with relatively large lasso coefficients are highlighted in green.
(TIF)

## Acknowledgments

We thank Balasubramanian Narasimhan for helpful discussion on the package development, Kenneth Tay and the members of the Rivas lab for insightful feedback. We thank the SBayesR and GCTB authors for their invaluable feedback and help with our SBayesR experiments.

This research has been conducted using the UK Biobank Resource under application number 24983. We thank all the participants in the study. The primary and processed data used to generate the analyses presented here are available in the UK Biobank access management system (https://amsportal.ukbiobank.ac.uk/) for application 24983, "Generating effective therapeutic hypotheses from genomic and hospital linkage data" (http://www.ukbiobank.ac.uk/wp-content/uploads/2017/06/24983-Dr-Manuel-Rivas.pdf), and the results are displayed in the Global Biobank Engine (https://biobankengine.stanford.edu).

Some of the computing for this project was performed on the Sherlock cluster. We would like to thank Stanford University and the Stanford Research Computing Center for providing computational resources and support that contributed to these research results.

## Author Contributions

**Conceptualization:** Robert Tibshirani, Manuel A. Rivas, Trevor Hastie.

**Data curation:** Yosuke Tanigawa, Matthew Aguirre, Manuel A. Rivas.

**Formal analysis:** Junyang Qian, Yosuke Tanigawa, Robert Tibshirani, Manuel A. Rivas, Trevor Hastie.

**Funding acquisition:** Robert Tibshirani, Manuel A. Rivas, Trevor Hastie.

**Investigation:** Junyang Qian, Yosuke Tanigawa, Wenfei Du, Robert Tibshirani, Manuel A. Rivas, Trevor Hastie.

**Methodology:** Junyang Qian, Robert Tibshirani, Manuel A. Rivas, Trevor Hastie.

**Project administration:** Robert Tibshirani, Manuel A. Rivas, Trevor Hastie.

**Resources:** Robert Tibshirani, Manuel A. Rivas, Trevor Hastie.

**Software:** Junyang Qian, Yosuke Tanigawa, Chris Chang.

**Supervision:** Robert Tibshirani, Manuel A. Rivas, Trevor Hastie.

**Validation:** Yosuke Tanigawa, Robert Tibshirani, Manuel A. Rivas, Trevor Hastie.

**Visualization:** Junyang Qian, Yosuke Tanigawa, Wenfei Du.

**Writing – original draft:** Junyang Qian, Wenfei Du, Trevor Hastie.

**Writing – review & editing:** Junyang Qian, Yosuke Tanigawa, Wenfei Du, Matthew Aguirre, Robert Tibshirani, Manuel A. Rivas, Trevor Hastie.

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
