## [Decision Letter · Decision Letter 0]

20 Mar 2020

Dear Dr Hastie,

Thank you very much for submitting your Research Article entitled 'A Fast and Scalable Framework for Large-scale and Ultrahigh-dimensional Multivariate Genome-wide Predictive Modeling with Application to the UK Biobank' to PLOS Genetics. Your manuscript was fully evaluated at the editorial level and by independent peer reviewers. The reviewers appreciated the attention to an important problem, but raised some substantial concerns about the current manuscript. Based on the reviews, we will not be able to accept this version of the manuscript, but we would be willing to review again a much-revised version. That is, we will consider a revised manuscript that robustly demonstrates marked improvement over the existing approaches (i.e. Bayesian Approaches, BLUP and polygenic risk scores), as the reviewer 1 pointed out.    We cannot, of course, promise publication at that time.

If you decide to revise the manuscript for further consideration at PLOS Genetics, please aim to resubmit within the next 60 days, unless it will take extra time to address the concerns of the reviewers, in which case we would appreciate an expected resubmission date by email to plosgenetics@plos.org.

[LINK]

We are sorry that we cannot be more positive about your manuscript at this stage. Please do not hesitate to contact us if you have any concerns or questions.

Yours sincerely,

Xiaofeng Zhu

Associate Editor

PLOS Genetics

Gregory P. Copenhaver

Editor-in-Chief

PLOS Genetics

Reviewer's Responses to Questions

**Comments to the Authors:**

Reviewer #1: Review of Qian et al

Summary:

This paper describes an efficient algorithm for fitting

the lasso regression model to large data sets, along with an implementation

(R package snpnet) and application to the UK biobank data to

obtain predictors (effectively performing genomic prediction, or

computing polygenic risk scores, PRS) for several different phenotypes.

The paper compares prediction accuracy with some other simpler methods.

The lasso is, in general, a widely studied, and also quite widely used method.

As such an algorithm and implementation for very large datasets are of potential interest

to a general audience both inside and outside genetics. However,

for readers of PloS genetics the interest is going to stand or fall on

the application: is Lasso a good method to do genomic prediction?

I am skeptical of this: the Lasso has never been the method of choice

for genomic prediction in smaller data sets, with the field generally

preferring other large-scale regression methods, including

very simple methods (eg "ridge regression", usually known

as BLUP in the quantitative genetics literature) or very computationally

intensive methods (Bayesian regression, usually fit via MCMC). The

Elastic Net is also sometimes used. But the Lasso, rarely.

While I will keep an open mind on whether this could change for

biobank-sized data, the current paper is unconvincing

on this because none of the comparisons are with state-of-the-art

approaches to this problem.

Overall then I think the main contribution of this paper

is the the algorithmic ideas, whose main appeal is their

simplicity and generality: I like the fact that the design allows

the algorithm to maximally exploit previous implementations, rather than

having to reimplement the coordinate ascent steps for example.

However, unless the resulting method is really competitive with state

of the art for genomic prediction then this seems better suited to

another journal.

Detailed comments:

1. Comparisons with other methods:

The methods used here do not seem to

represent a reasonable selection of state-of-the-art approaches to forming predictors for

genetic data, on which there is a large literature. Historically

genomic prediction has been done using multiple linear regression

fit either using very simple methods (eg "ridge regression", usually known

as BLUP in the quantitative genetics literature) or very computationally

intensive methods (Bayesian regression, usually fit via MCMC).

More recently, motivated by the difficulty of accessing/sharing

genotype data, as well as computational considerations,

a literature has sprung up around methods that attempt

to build predictors based on summary statistics only (and LD from

a reference panel). For example, Ge et al and Lloyd-Jones et al:

https://www.nature.com/articles/s41467-019-09718-5

https://www.nature.com/articles/s41467-019-12653-0

are recent examples, and includes comparisons with other methods,

the latter specifically on the UK biobank data with some of the same

phenotypes considered here.

To take a quick example, in Fig 2 of Lloyd Jones, looking at R2

for BMI, the performance among the methods they consider

ranges from 0.1 to 0.126. In this paper

(Table 3) Lasso achieves 0.103. I realize these numbers are not directly

comparable, being based on different protocols (CV splits etc)

for analyzing the UK biobank data, but it illustrates my concern that

Lasso may not be competitive with the best existing methods.

2. Algorithmic description

I found much of the algorithmic description in the overview overly long and hard to follow.

The basic idea seems rather simple (which is a good thing!) but the presentation seems

to obscure the simplicity rather than highlighting it.

The formal presentation of Algorithm 1 in section 4 helps a lot, and I suggest it should be

moved to the overview section. This should allow the text in the algorithmic

overview to be shortened, since much of the words seems to be

repeating, in less precise terms, what is given in Algorithm 1.

Also:

- the algorithm and text did not seem to address what happens if the "checking"

step fails. That is, in step 5 of Algorithm 1, what if no lambda satisfies

the KKT conditions? Or is this guaranteed not to happen?

- How is M chosen? Does it matter?

- the algorithm seems to rely on the fact that marginal screening is

going to be effective at identifying the correct variables to add in.

In some cases with complex correlations among variables this may not be true -

one can construct problems where the best pair of variables to include

are not among the marginally strongest. How does the algorithm

cope with that kind of situation? Is it guaranteed to

converge in practice?

3. Standardization

The question of whether or not to standardize variables is usually phrased in terms of modeling

assumptions -- if rare SNPs have bigger effects than common SNPs then standardization

could be appropriate and improve predictive performance. The paper suggests

that standardization will produce

worse performance but this is not obvious a priori - it should be shown empirically.

4. Implementation

The software implementation does not

appear to be quite ready for widespread distribution

(e.g. the R package on github has no man pages, and I could

not find a minimal working example).

Other

- the use of the term "multivariate" in the context of a multiple regression with univariate

outcome is rather confusing. From the title I expected the paper to deal with multivariate outcomes.

Better to stick to "multiple regression", or perhaps "multi-SNP regression" if you prefer.

- references to heritability were also confusing. E.g.

the abstract refers to "state-of-the-art heritability estimation",

when the goal here seems not to be heritability estimation but

building a predictor, which are different things.

Heritability provides an upper

bound on prediction accuracy from genetic data, but building a predictor is not

the same as "estimating heritability", and most approaches to estimating heritability

do not explicitly build predictors. I think you can (and probably should)

write the whole paper without mentioning heritability, and focussing entirely on PRS and prediction accuracy.

- the presentation of result is much longer than it need be. The main results for different

phenotypes and methods could probably be shown in a single figure (e.g. Lloyd-Jones Fig 2).

Many of the other figures did not seem essential to the main story.

Refs:

Ge, T., Chen, C., Ni, Y. et al. Polygenic prediction via Bayesian regression and continuous shrinkage priors. Nat Commun 10, 1776 (2019). https://doi.org/10.1038/s41467-019-09718-5

Lloyd-Jones, L.R., Zeng, J., Sidorenko, J. et al. Improved polygenic prediction by Bayesian multiple regression on summary statistics. Nat Commun 10, 5086 (2019). https://doi.org/10.1038/s41467-019-12653-0

Reviewer #2: How to build the best predictive model using large-scale genetic data is important in health and disease studies. This paper provides a true regression approach for this problem, an important alternative to the polygenic risk scores. The results from analysis of the UK Biobank are convincing and interesting. The algorithm seems to be quite reasonable.

I only have a few minor comments - (1) since Lasso results in biased estimates of the regression coefficients. Do the authors think that by performing further debased estimation, one can further improve the prediction performance? (2) since a very large number of SNPs are selection for each of the data examples, would the consistency results still hold? Lasso theory requires that the model has to be very sparse. (3) Why univariate screening + Lasso does not perform as well as fitting Lasso using all the SNPs? Does this mean that the univariate screening as proposed by Jianqin Fan etc does not really work in the settings considered in this paper?

**Have all data underlying the figures and results presented in the manuscript been provided?**

Reviewer #1: None

Reviewer #2: Yes

PLOS authors have the option to publish the peer review history of their article (what does this mean?). If published, this will include your full peer review and any attached files.

Reviewer #1: Yes: Matthew Stephens

Reviewer #2: No

---

## [Decision Letter · Decision Letter 1]

13 Jul 2020

Dear Dr Hastie,

Thank you very much for submitting your Research Article entitled 'A Fast and Scalable Framework for Large-scale and Ultrahigh-dimensional Sparse Regression with Application to the UK Biobank' to PLOS Genetics. Your manuscript was fully evaluated at the editorial level and by independent peer reviewers. The reviewers appreciated the attention to an important topic but identified some aspects of the manuscript that should be improved.

We therefore ask you to modify the manuscript according to the review recommendations before we can consider your manuscript for acceptance. Your revisions should address the specific points made by reviewers. Reviewer #1 raised important issues regarding the results of SBayesR.  This issue will need to be fully resolved in the revision and the editors agree that it may be very helpful for you to reach out the authors of SBayesR during your revision process, but we leave that up to you to decide.

[LINK]

Yours sincerely,

Xiaofeng Zhu

Associate Editor

PLOS Genetics

Gregory P. Copenhaver

Editor-in-Chief

PLOS Genetics

Reviewer's Responses to Questions

**Comments to the Authors:**

Reviewer #1: The authors have done a thorough job responding to my comments, and I believe the whole paper is much improved.

Just one new substantive issue has arisen during this revision: the results reported for the SBayesR method

are very poor, and seem to strongly contradict the original publication on this method. Indeed, it is a bit

hard to believe that it performs quite so poorly, and the reasons for its poor performance need to be understood

and either corrected or explained. For example, for height, SBayesR does no better than just Age + Sex in predicting

height - so it essentially has a 0% R2 when you consider the genetic component only. In contrast, LLoyd-Jones et al

report that SBayesR achieved an R2 of >35% for height in the UK biobank.

Something is clearly wrong, either with the SBayes software or with

the way it has been applied. (Other traits show a similar pattern, but the height result is particularly striking.)

Of course, I don't know what the problem is, but I suggest a first step would be to ask the SBayesR authors

if they have suggestions, and/or get their original code and see if you can reproduce their published results.

Other items:

in SBayes i noticed you excluded the MHC. Maybe this is recommended by SBayes software, but it seems likely to hurt R2 and

AUC for many traits as the MHC has a strong effect on many traits.

To make results comparable across methods it seems necessary to either exclude or include MHC for all methods.

(It seems unlikely that this issue explains the poor performance on height noted above.)

Reviewer #2: my previous comments were minor and the authors have addressed these comments.

**Have all data underlying the figures and results presented in the manuscript been provided?**

Reviewer #1: **No: **UK Biobank data can't be provided

Reviewer #2: Yes

PLOS authors have the option to publish the peer review history of their article (what does this mean?). If published, this will include your full peer review and any attached files.

Reviewer #1: **Yes: **Matthew Stephens

Reviewer #2: No

---

## [Editor Report · Decision Letter 2]

4 Sep 2020

Dear Dr Hastie,

We are pleased to inform you that your manuscript entitled "A Fast and Scalable Framework for Large-scale and Ultrahigh-dimensional Sparse Regression with Application to the UK Biobank" has been editorially accepted for publication in PLOS Genetics. Congratulations!

Yours sincerely,

Xiaofeng Zhu

Associate Editor

PLOS Genetics

Gregory P. Copenhaver

Editor-in-Chief

PLOS Genetics

Comments from the reviewers (if applicable):

**Data Deposition**

http://datadryad.org/submit?journalID=pgenetics&manu=PGENETICS-D-20-00068R2

**Press Queries**

---

## [Editor Report · Acceptance letter]

13 Oct 2020

PGENETICS-D-20-00068R2 

A Fast and Scalable Framework for Large-scale and Ultrahigh-dimensional Sparse Regression with Application to the UK Biobank 

Dear Dr Hastie, 

We are pleased to inform you that your manuscript entitled "A Fast and Scalable Framework for Large-scale and Ultrahigh-dimensional Sparse Regression with Application to the UK Biobank" has been formally accepted for publication in PLOS Genetics! Your manuscript is now with our production department and you will be notified of the publication date in due course.

With kind regards,

Matt Lyles

PLOS Genetics

On behalf of:
